# Estimating Multi-cause Treatment Effects via Single-cause Perturbation

**Zhaozhi Qian**
University of Cambridge
zhaozhi.qian@maths.cam.ac.uk

**Alicia Curth**
University of Cambridge
amc253@cam.ac.uk

**Mihaela van der Schaar**
University of Cambridge
UCLA
The Alan Turing Institute
mv472@cam.ac.uk

## Abstract

Most existing methods for conditional average treatment effect estimation are designed to estimate the effect of a *single cause* — only one variable can be intervened on at one time. However, many applications involve simultaneous intervention on multiple variables, which leads to *multi-cause* treatment effect problems. The multi-cause problem is challenging because one needs to overcome the confounding bias for a large number of treatment groups, each with a different cause combination. The combinatorial nature of the problem also leads to severe data scarcity — we only observe one factual outcome out of many potential outcomes. In this work, we propose Single-cause Perturbation (SCP), a novel two-step procedure to estimate the multi-cause treatment effect. SCP starts by augmenting the observational dataset with the estimated potential outcomes under single-cause interventions. It then performs covariate adjustment on the augmented dataset to obtain the estimator. SCP is agnostic to the exact choice of algorithm in either step. We show formally that the procedure is valid under standard assumptions in causal inference. We demonstrate the performance gain of SCP on extensive synthetic and semi-synthetic experiments.

## 1 Introduction

Estimating treatment effects from *observational data* is a central problem in causal inference and has many applications such as precision medicine [11]. In this work, we focus on estimating *conditional average treatment effects* (CATE) to reflect the heterogeneity within a population [1]. The vast majority of the CATE estimation methods consider the *single-cause* setting, where only *one* variable can be intervened on, e.g. the decision to give (or not to give) a particular drug. However, in many applications it is necessary to intervene on *multiple* variables simultaneously to achieve the desired outcome (the *multi-cause* setting). For example, multiple drugs are needed to treat patients with comorbid chronic diseases or systemic diseases such as cancer [21]. However, finding the best drug combination for each patient is very challenging and the current clinical practice is clearly sub-optimal [30]; studies have shown that nearly 50% of the elderly population in developed countries take one or more drugs that are *not* medically necessary [39]. Similar examples are abundant in the medical literature and beyond (Appendix A.5), which calls for a new methodology to estimate the combined effect of multiple causes (drugs), a challenge we undertake in this work.

We make a distinction between the terminology *cause* and *treatment*. We refer to a cause as an atomic variable that can be intervened on, and a treatment as a configuration of all causes. Therefore, if the problem involves $K$ causes and each cause is a binary variable, there will be $2^K$ possible treatments. The key objective of causal inference is to overcome the confounding bias in treatment assignment. This is challenging in the multi-cause setting because a large number of treatment groups need to be

35th Conference on Neural Information Processing Systems (NeurIPS 2021).

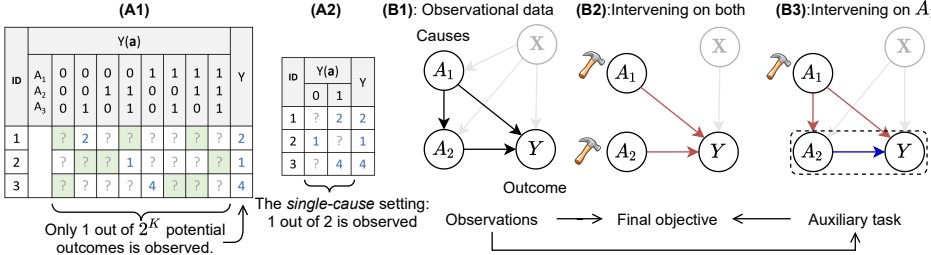

Figure 1: **(A) Illustration of the data scarcity challenge**. A1: $K = 3$ causes and A2: the single-cause setting. Each row contains one observation. Three green cells in each row will be filled in by SCP's first step to form the augmented dataset. **(B) Interventions on an illustrative DAG**. B1: observational data (no intervention), B2: intervening on both causes, B3: intervening on $A_1$ only. In B3, the intervention on $A_1$ generates an effect on the outcome and the cause $A_2$. The covariate $\mathbf{X}$ is greyed out for visual clarity.

balanced. (In contrast, one only needs to balance *two* treatment groups in the single-cause setting.) Furthermore, the combinatorial treatment also leads to *data scarcity* — we can only observe the outcome under the treatment that was given (factual outcome), but not the potential outcomes (PO) under all other treatments ($2^K - 1$ in total, as illustrated in Figure 1 A). As the number of causes increases, the fraction of observed outcomes decreases exponentially, which challenges the reliable estimation of CATE.

Most single-cause methods consider only two treatments (treated or untreated). In fact, many popular architectures and regularization methods aimed to overcome confounding do not scale computationally to large treatment spaces [57, 73, 58, 38]. As a remedy, one may make additional assumptions on the data generating process (DGP), for instance, assuming a linear model generates the outcome [29] or a low-dimensional latent variable generates the treatment [75]. However, such assumptions may limit the scope of application.

In this work, we take a different direction: instead of making additional assumptions on the functional form or latent variable structure, we exploit the connection between a single-cause intervention and a multi-cause intervention (Figure 1 B1-3). We establish that, under standard assumptions in causal inference, the single and multi-cause potential outcomes are equal in expectation under appropriate conditioning.

Based on this finding, we propose single-cause perturbation (SCP), a novel *two-step* procedure to estimate CATE in the multi-cause setting. In the first step, SCP generates $K$ additional datasets by predicting the potential outcomes resulting from perturbing each of the $K$ causes to their opposite value. It then performs covariate adjustment on the combined dataset. By deliberately perturbing the causes, the treatment assignment in the augmented dataset would be more *balanced* than the observational data, thereby reducing the confounding bias. SCP is agnostic to the exact choice of algorithm in either step and the user can choose the algorithms based on the application.

**Contributions**. We present SCP, a two-step multi-cause CATE estimator that leverages the connection between single and multi-cause interventions. SCP overcomes confounding bias by using single-cause CATE estimators to augment the observational data with the estimated potential outcomes. Compared with existing works, SCP does not make assumptions about the distributional or functional form of the DGP, making it suitable for complex problems in healthcare. We demonstrate and analyze the performance gain of SCP via extensive experiments.

## 2 Problem formulation and notations

In this work, we focus on the CATE estimation problem with $K$ binary causes.[1] Let the causes $\mathbf{A} = (A_1, \ldots, A_K)$ be a multi-dimensional random variable with sample space $\Omega = \{0, 1\}^K$, where $A_k$ is the $k^{\text{th}}$ cause. Let $\mathbf{A}_{-k} \in \Omega_{-k} = \{0, 1\}^{K-1}$ be the collection of all but the $k^{\text{th}}$ cause. Let $\mathbf{X} \in \mathbb{R}^D$ and $Y \in \mathbb{R}$ be the covariates and observed outcomes respectively. The causal relationship

---

[1]SCP also applies to multi-level categorical causes, i.e. $A_k \in \{0, 1, \ldots, L\}$, $L \in \mathbb{N}^+$ and multi-dimensional outcomes, i.e. $Y \in \mathbb{R}^M$. Here, we use the current setting for illustration.

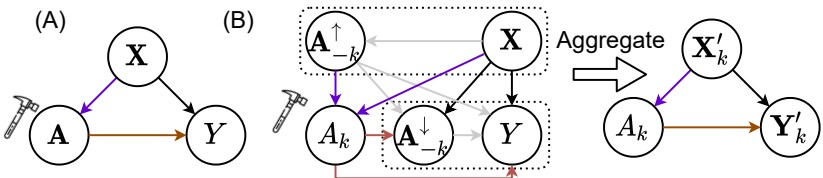

Figure 2: Illustrative causal graphs. (A) Intervention on all causes $\mathbf{A}$. (B) Intervention on the single cause $A_k$. The other causes are partitioned into descendants $\mathbf{A}^{\downarrow}_{-k}$ and non-descendants $\mathbf{A}^{\uparrow}_{-k}$. Purple edges: confounding to treatment assignment. Brown edges: effects on the (combined) outcomes. Some less important edges are greyed out for visual clarity.

between these variables is illustrated in Figure 2 A, which is a direct generalization of the single cause setting [56]. We have access to an observational dataset $\mathcal{D}_0 = \{\mathbf{x}_i, y_i, \mathbf{a}_i\}_{i \in [N_0]}$ with $N_0$ independent samples from the random variables defined above. Throughout the text we use capital letters for random variables and lower case letters for fixed constants. We use boldface for vectors or multi-dimensional random variables. When the context is clear, we will simplify the conditional expressions, e.g. $\mathbb{P}(Y|\mathbf{X}) := \mathbb{P}(Y|\mathbf{X} = \mathbf{x})$.

## 2.1 Multi-cause intervention

We formulate the CATE estimation problem using the potential outcome (PO) framework [56].[2] Let $Y(\mathbf{a}) \in \mathbb{R}$ denote the potential outcome in a world where the treatment $\mathbf{a} \in \Omega$ was given. We would like to estimate the CATE between any two treatments given the covariates i.e. $\tau(\mathbf{a}, \mathbf{a}', \mathbf{x}) = \mathbb{E}[Y(\mathbf{a}) - Y(\mathbf{a}')|\mathbf{X} = \mathbf{x}], \forall \mathbf{a}, \mathbf{a}' \in \Omega, \mathbf{x} \in \mathbb{R}^D$. We can estimate CATE by estimating all potential outcomes $\mathbb{E}[Y(\mathbf{a})|\mathbf{X}], \forall \mathbf{a} \in \Omega$.

The following three assumptions have been proposed to identify the multi-cause PO [56, 24]. (1) *Consistency*: $\forall \mathbf{a} \in \Omega$ if $\mathbf{A} = \mathbf{a}, Y(\mathbf{a}) = Y$. (2) *Weak unconfoundedness*: $Y(\mathbf{a}) \perp\!\!\!\perp \mathbf{A} \mid \mathbf{X}, \forall \mathbf{a} \in \Omega$. (3) *Overlap*: $\mathbb{P}(\mathbf{A} = \mathbf{a}|\mathbf{X}) > 0, \forall \mathbf{a} \in \Omega$, if $\mathbb{P}(\mathbf{X}) > 0$. The assumptions stated above allow the expectation of multi-cause PO to be estimated from observational data: $\forall \mathbf{a} \in \Omega, \forall \mathbf{x} \in \mathbb{R}^D$:

$$\mathbb{E}[Y(\mathbf{a})|\mathbf{X} = \mathbf{x}] = \mathbb{E}[Y|\mathbf{X} = \mathbf{x}, \mathbf{A} = \mathbf{a}] \tag{1}$$

## 2.2 Single-cause intervention

Here we consider the intervention on a single-cause, e.g. adding a new drug $A_1$ to the existing medications. Such intervention may affect the outcome and the other causes. For example, the inclusion of drug $A_1$ may promote the usage of another drug $A_2$ because $A_2$ can mitigate the side effects of $A_1$ [48].

We denote $Y(a_k) \in \mathbb{R}$ as the potential outcome where the cause $A_k$ is set to be $a_k$. We refer to $Y(a_k)$ as the **single-cause PO**. Note that the single-cause PO $Y(a_k)$ is different from the multi-cause PO $Y(\mathbf{a})$ because the latter refers to a potential world where *all* causes are intervened on. We sometimes denote the multi-cause PO as $Y(\mathbf{a}) := Y(a_k, \mathbf{a}_{-k})$.

We assume that, based on domain knowledge, we can partition the rest of the causes $\mathbf{A}_{-k}$ into $A_k$'s causal descendants $\mathbf{A}^{\downarrow}_{-k}$ and its non-descendants $\mathbf{A}^{\uparrow}_{-k}$ as illustrated in Figure 2 B [45]. We denote $\mathbf{A}_{-k}(a_k), \mathbf{A}^{\downarrow}_{-k}(a_k)$ and $\mathbf{A}^{\uparrow}_{-k}(a_k)$ as their potential outcomes respectively. By definition, the non-descendants should be unaffected by the intervention:

$$\mathbf{A}^{\uparrow}_{-k}(0) = \mathbf{A}^{\uparrow}_{-k}(1) = \mathbf{A}^{\uparrow}_{-k}. \tag{2}$$

As shown in Figure 2 B, it is convenient to aggregate all the variables affected by $A_k$ into a combined outcome $\mathbf{Y}'_k$, and aggregate all the variables confounding $A_k$ as a combined confounder $\mathbf{X}'_k$:

$$\mathbf{Y}'_k := (Y, \mathbf{A}^{\downarrow}_{-k}); \quad \mathbf{Y}'_k(a_k) := (Y(a_k), \mathbf{A}^{\downarrow}_{-k}(a_k)); \quad \mathbf{X}'_k := (\mathbf{X}, \mathbf{A}^{\uparrow}_{-k}) \tag{3}$$

---

[2]In Appendix A.4, we present an alternative formalism using do-operation [46]. We show that the same SCP algorithm can be derived using either formalism.

Table 1: Summary of the data augmentation task in SCP's first step.

| Equation | Target | Input Covariates | Estimated Value | Algorithm |
|----------|--------|------------------|-----------------|-----------|
| Eq. 2 | $\mathbf{A}^{\uparrow}_{-k}(a'_k)$ | - | $\mathbf{a}^{\uparrow}_{-k}(a'_k) = \mathbf{a}^{\uparrow}_{-k}$ | - |
| Eq. 4 | $\mathbf{A}^{\downarrow}_{-k}(a'_k)$ | $\mathbf{X}'_k$ | $\mathbf{a}^{\downarrow}_{-k}(a'_k) \sim \mathbb{P}(\mathbf{A}^{\downarrow}_{-k}|\mathbf{X}'_k, A_k)$ | DR-CFR |
| Eq. 4 | $Y(a'_k)$ | $\mathbf{X}'_k, \mathbf{A}^{\downarrow}_{-k}$ | $y(a'_k) = \mathbb{E}(Y|\mathbf{X}'_k, \mathbf{A}^{\downarrow}_{-k}, A_k)$ | DR-CFR |

To identify the combined PO $\mathbf{Y}'_k(a_k)$, we make the standard assumptions using $A_k$, $\mathbf{Y}'_k$, and $\mathbf{X}'_k$: (4) *Single-cause Consistency*: $\forall k \leq K$, $\forall a \in \{0,1\}$ if $A_k = a_k$, $\mathbf{Y}'_k(a_k) = \mathbf{Y}'_k$. (5) *Single-cause Unconfoundedness*: $\mathbf{Y}'_k(a_k) \perp\!\!\!\perp A_k \mid \mathbf{X}'_k, \forall a_k \in \{0,1\}, \forall k \leq K$. The multi-cause overlap (Section 2.1) implies single-cause overlap, but the multi-cause consistency and unconfoundedness *do not* imply the single-cause counterparts (Appendix A.3). Appendix A.1 Proposition 2 shows that, under these assumptions, we can identify $\mathbf{Y}'_k(a_k)$ from observational data as: $\forall k \leq K$, $\forall a_k \in \{0,1\}$,

$$\mathbb{P}(\mathbf{Y}'_k(a_k)|\mathbf{X}'_k) = \mathbb{P}(\mathbf{A}^{\downarrow}_{-k}|\mathbf{X}'_k, A_k = a_k) \cdot \mathbb{P}(Y|\mathbf{X}'_k, \mathbf{A}^{\downarrow}_{-k}, A_k = a_k). \tag{4}$$

**Discussion on partitioning the causes**. We can always partition the causes into descendants and non-decadents as long as the structure between the causes follows a DAG (hence no cycles). In practice, such structural knowledge is often available, e.g. we can use the clinical guidelines to identify the drugs whose prescription will be influenced by the usage of another drug. Note that we do not need to specify the causal graph of all individual variables (e.g. the link between two covariates $X_i$, $X_j$). However, when the full causal graph is available, we can adapt SCP to make use of the additional structural knowledge as discussed in Appendix A.6. On the other hand, we show empirically that SCP is not sensitive to misspecified partitioning (Section 5.1). Appendix A.3 contains an extended discussion on all our assumptions.

## 3 Single Cause Perturbation

### 3.1 The algorithm

In this section, we introduce our proposed method – single cause perturbation (SCP). Given an observational dataset $\mathcal{D}_0$ with $N_0$ data points: $\mathcal{D}_0 = \{\mathbf{x}_i, y_i, \mathbf{a}_i\}_{i \in [N_0]}$, SCP proceeds in two steps: it first fits a set of models that can predict the effects of changing *a single* cause, and uses them to create $K$ additional data sets $\mathcal{D}_k = \{\mathbf{x}_i, \tilde{y}^k_i, \tilde{\mathbf{a}}^k_i\}^{N_0}_{i=1}$, for $k \in [K]$, each corresponding to the potential scenario of perturbing a single cause. It then fits a final model on this enlarged dataset, which is used to estimate the multi-cause CATE. The pseudocode is detailed in Appendix A.7 Algorithm 1.

**Training single-cause models**. Based on Equation 4, we will train two separate models to estimate the combined PO $\mathbf{Y}'_k(a_k)$: one for $\mathbf{A}^{\downarrow}_{-k}(a_k)$ and one for $Y(a_k)$. Note that for CATE estimation, we only need to estimate the *expectation* $\mathbb{E}(Y|\mathbf{X}'_k, \mathbf{A}^{\downarrow}_{-k}, A_k)$ rather than the full probability distribution. The models are trained on the observational data $\mathcal{D}_0$. We can use any single-cause CATE estimator for this purpose since only one cause is intervened on.

We choose to use the state of the art single-cause CATE estimator, Disentangled Representations for Counterfactual Regression algorithm (DR-CFR) [22]. DR-CFR achieves higher estimation accuracy by learning to distinguish between true confounders, adjustment variables and instruments contained in $\mathbf{X}'_k$. We provide a self-contained description of DR-CFR in Appendix A.8.

**Data augmentation**. As illustrated in Table 1, once the single-cause models are fitted, sampling perturbed data points from observations $(\mathbf{x}, y, \mathbf{a}) \in \mathcal{D}_0$ involves three steps: (1) obtain $\mathbf{a}^{\uparrow}_{-k}(a'_k)$ directly from the observations, (2) sample $\mathbf{a}^{\downarrow}_{-k}(a'_k) \sim \mathbb{P}(\mathbf{A}^{\downarrow}_{-k}(a'_k)|\mathbf{X}'_k)$, and (3) obtain $y(a'_k) := \mathbb{E}(Y(a'_k)|\mathbf{X}'_k, \mathbf{A}_{-k}(a'_k))$. Here $a'_k = 1 - a_k$ corresponds to perturbing the cause $A_k$ (recall that $a_k \in \{0,1\}$). Note that in step two we sample the new causes $\mathbf{a}^{\downarrow}_{-k}(a'_k)$ from the distribution in order to keep them as binary variables. To generate a new data point $(\mathbf{x}, \tilde{y}^k, \tilde{\mathbf{a}}^k)$, we define $\tilde{y}^k := y(a'_k)$ and $\tilde{\mathbf{a}}^k := (a'_k, \mathbf{a}_{-k}(a'_k))$. Denote $\mathcal{D}_k = \{\mathbf{x}_i, \tilde{y}^k_i, \tilde{\mathbf{a}}^k_i\}^{N_0}_{i=1}$ as the perturbed data for $A_k$. We combine all perturbed datasets $\mathcal{D}_k$, $k \in [K]$ and the original dataset $\mathcal{D}_0$ to create the augmented training data

$\mathcal{D}^{Tr} = \{\mathcal{D}_k\}_{k \in [0,K]}$. For each unique $\mathbf{x}$, $\mathcal{D}^{Tr}$ contains $K+1$ different treatments $\mathbf{a}, \tilde{\mathbf{a}}^k, \ldots, \tilde{\mathbf{a}}^K$ and their corresponding outcomes.

**Covariate adjustment on augmented data**. We can estimate CATE by learning the conditional expectation in Equation 1 using the augmented data $\mathcal{D}^{Tr}$. We use a standard feed-forward neural network, $f_\theta : \mathbb{R}^D \times \Omega \to \mathbb{R}$ with trainable weights $\theta$.

## 3.2 Validity of SCP: linking single and multi-cause PO

One may wonder why the augmented data points (single-cause POs) would help estimate the multi-cause PO: they correspond to different interventions, i.e. intervention on a single cause versus intervention on all causes simultaneously. Proposition 1 shows that given our assumptions the single and multi-cause POs are equal in expectation under appropriate conditioning – therefore, (imputed) single cause POs can be used for multi-cause estimation. The proof is shown in A.1.

**Proposition 1** (Equivalence of the single and multi-cause PO's conditional expectation). *Under the sequential ignorability assumption [53], $\forall k \leq K$,*

$$\mathbb{E}(Y(a_k, \mathbf{a}_{-k})|\mathbf{X}) = \mathbb{E}(Y(a_k)|\mathbf{X}, \mathbf{A}_{-k}(a_k) = \mathbf{a}_{-k}). \tag{5}$$

Note that the $Y(a_k)$ and $\mathbf{A}_{-k}(a_k)$ on the right hand side (RHS) is precisely what we estimated and added to the augmented dataset $\mathcal{D}_k$ in the first step. Thus if we train a supervised learning model on $\mathcal{D}_k$ to estimate the RHS, the trained model can also estimate the multi-cause PO on the LHS. Moreover, since the relationship in Equation 5 holds for all $k$, we can pool all the augmented datasets into one training dataset $\mathcal{D}^{Tr}$, which is $K+1$ times the size of the observational data i.e. $|\mathcal{D}^{Tr}| = (K+1)|\mathcal{D}_0|$. The increased sample size mitigates the data scarcity issue and allows the estimator to generalize better.

Proposition 1 also highlights the necessity of estimating $\mathbf{A}_{-k}(a_k)$ in addition to $Y(a_k)$ in the first step. This is because Equation 5 is conditioned on $\mathbf{A}_{-k}(a_k)$ rather than the observed cause $\mathbf{A}_{-k}$. Note that $\mathbf{A}_{-k}(a_k) = \mathbf{A}_{-k}, \forall a_k \in \{0, 1\}$ only when $A_k$ has no descendants.

## 3.3 SCP creates a more balanced dataset via data augmentation

In addition to increased sample size, there is also a less obvious (but equally important) reason why SCP would achieve performance gain: the augmented data tend to be more *balanced* than the observational data. This is because SCP perturbs every single cause of all the observations. For instance, by combining $\mathcal{D}_0$ and $\mathcal{D}_1$, the empirical distribution $\hat{\mathbb{P}}(A_1|X = \mathbf{x}_i) = 0.5, \forall \mathbf{x}_i \in \mathcal{D}_0$. Balancing is important because prior research has shown that CATE estimators trained on a balanced dataset tend to generalize better [57]. In fact, many existing causal inference methods employ balancing techniques to improve performance (see Section 4). In Section 5.1, we demonstrate experimentally that SCP consistently improves the balancing of the observational dataset.

## 3.4 Trade off between sample size, balancing, and first step error

SCP's data augmentation increases sample size and improves balancing, both of which are beneficial to CATE estimation. However, there is a caveat: the augmented dataset will also carry the finite-sample estimation error made in the first step. There is a risk that this additional source of noise will reduce or even cancel out the benefits of data augmentation.

In the simulation study in Section 5.1 we investigate this empirically, and observe that SCP's actual error in the first step is usually much smaller than the error required to offset the benefits of data augmentation. We conjecture that this is because SCP only perturbs *one* cause at a time. The effect of such a localized perturbation can be efficiently estimated by the existing methods tailored for the single-cause setting.

One can envision an alternative way where we bundle together any two (or even more) causes $A_j$ and $A_k$ and perturb both of them simultaneously. This will further increase the sample size and improve the balancing, but the first step error will also increase because the effect of a joint perturbation is harder to estimate. After all, if we were able to do this well, there is no need for data augmentation in the first place.

Table 2: Comparison with the related works. The ATE methods are listed for completeness.

| Method | Ref | Estimand | Balancing method | Sample size | Intermediate estimand |
|---|---|---|---|---|---|
| SCP | This work | CATE | Data augmentation | $\uparrow\uparrow$ | $\mathbb{E}(\mathbf{Y}'_k(a'_k)\|\mathbf{X}')$ |
| Cov. Adjustment | [32] | CATE | None | $=$ | $\mathbb{P}(\mathbf{Y}\|\mathbf{X}, \mathbf{A})$ |
| VSR | [75] | CATE | Weighting | $=$ | $\mathbb{P}(\mathbf{A}\|\mathbf{Z}), \mathbb{P}(\mathbf{Z}\|\mathbf{X})$ |
| Deconfounder | [72] | CATE | None | $=$ | $\mathbb{P}(\mathbf{Y}\|\mathbf{Z}, \mathbf{A})$ |
| Weighting | [34] | ATE | Weighting | $=$ | $\mathbb{P}(\mathbf{A}\|\mathbf{X})$ |
| Matching | [37] | ATE | Matching | $\downarrow\downarrow$ | $\mathbb{P}(\mathbf{A}\|\mathbf{X})$ |
| G computation | [54] | ATE | Marginalization | NA | $\mathbb{P}(\mathbf{Y}\|\mathbf{X}, \mathbf{A})$ |

A complete theoretical analysis of the trade off is challenging because all three interacting factors contribute to the overall estimation error. Moreover, an important feature of SCP is that it does not make *any* assumption about the DGP (functional form or error distribution). However, such assumptions are usually necessary to establish statistical efficiency bounds [44]. For these reasons, we will defer the theoretical analysis of the trade off to future works.

# 4 Related works

## 4.1 Multi-cause and single-cause CATE estimation

Table 2 summarizes the causal inference methods related to SCP. The *covariate adjustment* method uses supervised learning to estimate the PO from the "feature vector" $(\mathbf{x}, \mathbf{a})$ by Equation 1 [60, 26].

In the single-cause setting, recent works have proposed various architectures and regularization methods [57, 38, 2, 73, 58, 74, 22]. Unfortunately, these methods often fail to scale with the number of treatments. For instance, the popular multi-head neural network architecture requires one output head for each of the $2^K$ treatment levels [57], which will be infeasible even with moderate-sized $K$.

In the multi-cause setting, Variational Sample Re-weighting (*VSR*) [75] and *Deconfounder* [72] improve estimation accuracy under additional assumptions about the DGP. Both methods assume that the propensity score (PS) is determined by low-dimensional latent variables $\mathbf{Z}$, i.e. $\mathbb{P}(\mathbf{A}|\mathbf{X}) = \sum_{\mathbf{Z}} \mathbb{P}(\mathbf{A}|\mathbf{Z})\mathbb{P}(\mathbf{Z}|\mathbf{X})$. This assumption also makes Deconfounder robust to a certain type of hidden confounders [72]. In comparison, SCP does not make this assumption and it improves balancing by data augmentation as discussed in Section 3.3. Other existing methods assume the outcome is generated from a linear model [43]; SCP does not make any assumption on the function form.

## 4.2 Multi-cause *average* treatment effect (ATE) estimation

The methods for multi-cause ATE estimation broadly fall into two categories: *weighting* and *matching* [25, 37]. The weighting methods assign an importance weight to each data point in order to create a balanced dataset for ATE estimation [15, 34]. To adapt these methods for CATE estimation, we could perform covariate adjustment on the weighted data. In comparison, matching methods achieve balancing by removing unmatched data points and will end up with a smaller dataset [37, 7, 62]. Since CATE is a much more complex estimand than ATE (and thus requires more samples), matching methods designed for ATE are unlikely to achieve good performance for multi-cause CATE estimation.

G-Computation is also a technique for ATE estimation [54, 8]. To compute the average effect, G-computation marginalizes over the confounders $\mathbf{X}$. The standard implementation estimates the covariate distribution $\mathbb{P}(\mathbf{X})$ and uses Monte Carlo sampling for marginalization [52, 63]. This makes G-computation conceptually very different from SCP because SCP's data augmentation is unrelated to marginalization – its purpose is to increase sample size and balancing for covariate adjustment. We discuss several other less related works in Appendix A.9.

## 4.3 Causal data augmentation

Causal data augmentation uses known or learned causal structure to generate augmented datasets (in contrast to heuristic data augmentation [59, 36]). Several recent works apply this approach to domain

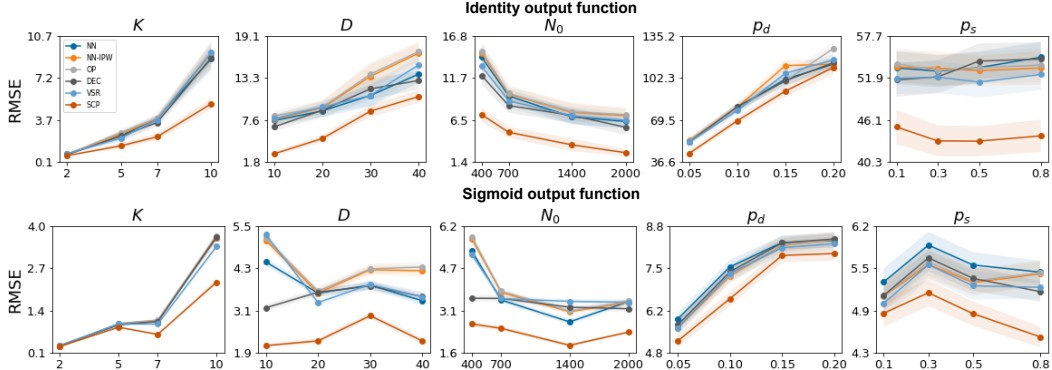

Figure 3: **Simulation Results** (best viewed in color). RMSE is plotted with the 95% confidence interval shaded (the lower the better). Algorithms include NN, NN-IPW, OP, DEC, VSR and SCP. CFR and DR-CFR's RMSE is an order of magnitude bigger and is shown in Appendix A.10 separately.

adaptation [64, 27], robustness [35, 65] and reinforcement learning [47]. To our knowledge, SCP is the first method that applies causal data augmentation to multi-cause CATE estimation.

# 5 Experiments

## 5.1 Simulation study

**Dataset**.[3] We created a range of synthetic datasets to examine the performance of SCP under different scenarios. Each dataset contains $N_0$ samples for training, 200 samples for validation and 4000 for testing. The training and validation sets contain observations $(\mathbf{x}_i, y_i, \mathbf{a}_i)$ whereas the testing set contains $(\mathbf{x}_i, y_i(\mathbf{a})), \forall \mathbf{a} \in \Omega$. To generate an observation, we first sample $D$ covariates independently: $\forall d \leq D, x_{id} \sim N(0, 1)$. Then we obtain the causes $a_{ik}, \forall k \leq K$ and the outcome $y_i$:

$$a_{ik} \sim \mathrm{B}\big[\sigma\big(\sum_{m=1}^{D} v_m x_{im} + \sum_{n=1}^{k-1} u_n a_{in} + \epsilon_{ik}\big)\big]; \quad y_i = \phi\big(\sum_{l=1}^{L} s_l x'_{il} + \sum_{l=1}^{L}\sum_{j=l}^{L} d_{lj} x'_{il} x'_{ij} + \varepsilon_i\big), \quad (6)$$

where $\mathbf{x}'_i = (\mathbf{x}_i, \mathbf{a}_i, 1) \in \mathbb{R}^L$, $v, u, s, d$ are weights, $\mathrm{B}[\cdot]$ denotes a Bernoulli random variable, $\sigma$ denotes the sigmoid function, $\phi$ is either identity or the sigmoid function depending on the simulation setting. To generate various response surfaces, only a fraction $p_s$ of the weights $s$ are non-zero and sampled i.i.d from $N(0, 1)$, resulting in not all covariates and causes contributing to the outcome. The weights $d$ are generated in the same way with the sparsity controlled by $p_d$, resulting in varying degrees of interaction between covariates and causes. The weights $v, u$'s are obtained similarly with sparsity $p_v = p_u = 0.3$. $\epsilon$ and $\varepsilon$ are white noises sampled from $N(0, 0.01)$. We evaluate the models using the Root Mean Squared Error (RMSE) on all potential outcomes, which is defined as $\sqrt{\frac{1}{N_t}\sum_{i=1}^{N_t}\sum_{\mathbf{a}_i \in \Omega}(\mathbb{E}[Y(\mathbf{a}_i)|X_i] - \hat{y}(\mathbf{a}_i))^2}$. The simulation parameters of all the experiments below are listed in Appendix A.10 Table 5.

**Benchmarks**. We included *seven* benchmarks to compare with SCP. As a baseline, we used covariate adjustment with feed-forward neural networks (**NN**). We compared with **VSR** and Deconfounder (**DEC**), the SOTA methods in multi-cause CATE estimation [75, 72]. For completeness, we also included Counterfactual Regression (**CFR**) and **DR-CFR** from the single-cause CATE literature [57, 22] as well as the propensity score (**NN-IPW**) and overlap score (**OP**) methods from the ATE literature [25, 34]. Appendix A.10 describes training and hyper-parameter tuning procedure in detail.

**Main results**. In total, we performed 168 simulations with different sets of parameters. The main results are presented in Figure 3 (additional results in Appendix A.12). In each panel, one simulation parameter is varied while the rest are fixed (see Appendix A.10). SCP consistently outperforms

---

[3]The implementation of SCP and the experiment code are available at `https://github.com/ZhaozhiQIAN/Single-Cause-Perturbation-NeurIPS-2021` or `https://github.com/orgs/vanderschaarlab/repositories`

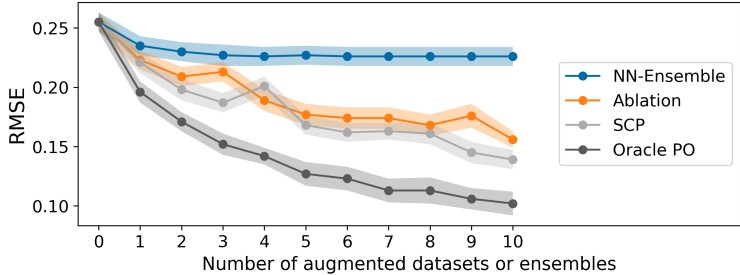

Figure 4: **The inclusion of augmented data points reduces error**. RMSE as more datasets $\mathcal{D}_k$ are added to $\mathcal{D}^{Tr}$ or more models are added to the NN ensemble. In total, there are $K = 10$ causes in this simulation.

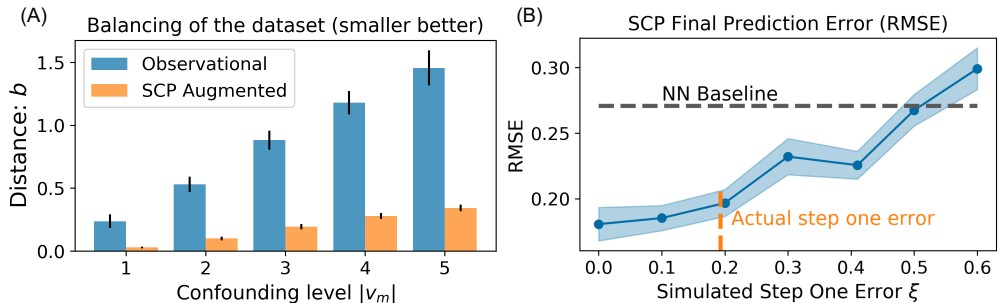

Figure 5: **(A): SCP consistently improves the balancing of the observational data.** Error bars represent the standard deviation of five runs. **(B): Relationship between the step one and the final prediction error.** A first step error of 0.4 will degrade SCP's overall performance to the NN baseline (dotted horizontal line). However, the actual step one error is only half of that value (around 0.2).

the benchmarks across different number of causes $K$, covariate dimensionality $D$, sample sizes $N_0$, and sparsity of the causal structure $p_s, p_d$. The performance gain becomes more pronounced as the number of causes increase, e.g. $K = 10$. Note that VSR and DEC's DGP assumption is approximately valid here because the $v_m$ and $u_n$ that govern treatment assignment are sparse vectors (Equation 6).

**Why is SCP working?** SCP's performance gain roots from the increase in sample size and the improvement in balancing. In Figure 4, we show that SCP's prediction accuracy improves consistently as each augmented dataset $\mathcal{D}_k, k \in [0, K]$ is added to the training data $\mathcal{D}^{Tr}$ (this simulation involves $K = 10$ causes). The benchmark **NN ensemble** refers to an ensemble of NN models trained using the bootstrapped observational data $\mathcal{D}_0$ [50]. The performance improvements of NN ensemble is much slower and smaller than SCP because it only bootstraps $\mathcal{D}_0$ without augmenting it with new data points. The other benchmarks in the figure will be discussed later.

To measure the improvements in balancing, we use the sum of the distributional distances between the treatment groups, i.e. $b = \sum_{\mathbf{a} \in \Omega} \text{MMD}(\mathbb{P}(\mathbf{X}|\mathbf{A} = \mathbf{a}), \mathbb{P}(\mathbf{X}|\mathbf{A} \neq \mathbf{a}))$, where MMD is the maximum mean discrepancy [4]. The value $b$ appears in the generalization bound of a CATE estimator [57] (also see Appendix A.2). Hence, achieving smaller $b$ (more balancing) is highly desirable. We generated a range of observational datasets with varying confounding levels, and use SCP to augment each dataset (the confounding level is controlled by the $v_m$ in Equation 6). Figure 5 (A) shows that SCP's augmented data is consistently more balanced than the observational data (the improvements in RMSE is shown in Appendix A.12).

**Relationship between step one error and overall error.** Next, we study how the step one error affects the overall error. We set the augmented data points to be the true expected PO corrupted by Gaussian noise: $\tilde{y}_k = \mathbb{E}(Y(a'_k)|\mathbf{X}'_k, \mathbf{A}^{\downarrow}_{-k}) + \xi$. The standard deviation of $\xi$ is a proxy for step one error. As expected, Figure 5 B shows that the overall error increases with the step one error. SCP's performance becomes similar to the NN baseline (black line) when the step one error reaches 0.4, which is twice as much as SCP's actual step one error 0.2 (dotted orange line).

Table 3: Results of the semi-synthetic data experiment using different data sizes $N_0$.

| Method | RMSE | | | Ranking Error | | |
|---|---|---|---|---|---|---|
| | $N_0 =500$ | 1000 | 1500 | $N_0 =500$ | 1000 | 1500 |
| NN | 1.257 (.004) | 1.383 (.006) | 1.116 (.004) | 282.3 (0.9) | 321.6 (1.0) | 228.1 (1.5) |
| VSR | 1.246 (.004) | 1.186 (.004) | 1.140 (.005) | 270.3 (1.2) | 253.4 (1.4) | 233.6 (1.6) |
| DEC | 1.268 (.004) | 1.200 (.004) | 1.118 (.005) | 283.9 (0.8) | 259.1 (1.3) | 236.4 (1.5) |
| CFR | 2.028 (.006) | 1.924 (.007) | 1.856 (.008) | 393.2 (1.0) | 380.8 (1.1) | 335.4 (1.3) |
| DR-CFR | 2.118 (.006) | 2.005 (.008) | 1.929 (.008) | 401.1 (1.0) | 391.2 (1.1) | 379.6 (1.4) |
| NN-IPW | 1.354 (.005) | 1.244 (.003) | 1.123 (.004) | 295.4 (0.8) | 253.0 (1.0) | 225.9 (1.4) |
| OP | 1.365 (.005) | 1.426 (.006) | 1.215 (.005) | 287.8 (0.8) | 316.1 (1.0) | 238.1 (1.4) |
| SCP | **1.117 (.004)** | **1.098 (.004)** | **1.044 (.004)** | **230.5 (1.3)** | **221.3 (1.4)** | **217.9 (1.4)** |

**Sensitivity to mis-specified partitioning and step one error**. To better understand the sensitivity, we compare the SCP with an ablated version (**Ablation**) where there is no prior knowledge about the non-descendants of a single cause, i.e. $\mathbf{A}^{\uparrow}_{-k} = \varnothing$. As a reference, we also consider **Oracle PO**, a SCP with error-free data augmentation step. Figure 4 shows that the correct partitioning of causes is indeed important because the ablation incurred noticeable performance loss compared with other SCP versions. However, even the ablated version consistently outperforms the ensemble of NN. This suggests that the increase in sample size and balancing tend to bring more benefit than the noise introduced in the first step. In fact, the Oracle PO achieves more than 60% performance improvement over the NN, which gives a wide "safety margin" for step one error.

**Further experiments.** In Appendix A.12, we present additional simulation studies that further illustrate SCP's source of performance gain under different settings. Our results consistently suggest that the increase in sample size and the improvement in balancing are the two key drivers of the gain.

## 5.2 Semi-synthetic experiment

**Dataset**. We used the de-identified COVID-19 Hospitalization in England Surveillance System (CHESS) data, which contains individual-level risk factors, treatments and outcomes of $N = 3,090$ ICU patients admitted during the first peak of the pandemic. Based on the prior research on COVID-19 [20, 49], we extracted $D = 17$ covariates $\mathbf{X}$ (e.g. age and multi-morbidity) and $K = 5$ causes $\mathbf{A}$ (e.g. ventilation and anti-viral treatments). The full list of covariates, causes and the assumed causal structure are shown in Appendix A.11. The outcome of interest is the patient's length of stay (LoS) in ICU [51]. Achieving shorter LoS is crucial for handling the large influx of patients during the peak of pandemic. We simulate the potential LoS for all treatments based on the state-of-the-art LoS model proposed in [70], which is a generalized linear model with interactions:

$$\log Y(\mathbf{a}) = \sum_{j,k \in [D+K+1]} \beta_{jk} x'_j x'_k + \xi, \tag{7}$$

where $\mathbf{x}' = (\mathbf{x}, \mathbf{a}, \mathbf{1})$ is the concatenation of the covariates, causes and a vector of ones, $\beta_{ij}$ is the coefficient sampled from $N(0, 0.5)$ and $\xi$ is white noise $N(0, 0.1)$.

**Training and evaluation**. We use the same benchmarks as in the simulation study. After sorting the data chronologically according to the date of admission, we train and tune the algorithms on the first $N_0$ patients, and perform evaluation on the rest of the patients. Compared with random splitting, this evaluation strategy preserves the temporality of the data and better mimics the actual training and deployment of the algorithm. For decision support, we would like the CATE estimator to rank higher the treatments that lead to better potential outcomes. Therefore, in addition to RMSE, we also report the ranking error, measured by the Spearman's Footrule distance between the treatment rankings induced by the true and the estimated POs [31]. A detailed explanation of the distance is given in Appendix A.11.

**Results**. The experimental results are presented in Table 3. We find that SCP consistently outperforms the benchmarks in both evaluation metrics. Achieving smaller ranking error means that SCP is better at creating a short list of plausible treatment plans for the clinicians to choose from. In practice, narrowing down the large number of treatments into a short list might help streamline the clinician's decision process and improve efficiency. Moreover, SCP also consistently achieves the best accuracy in terms of RMSE and its performance is relatively stable and improving when $N_0$ increases.

It is worth highlighting that SCP is more *data efficient* than the benchmarks: it achieves better RMSE with $N_0 = 500$ samples than the benchmarks trained with $N_0 = 1500$ samples. Being data efficient is crucial for urgent applications such as pandemic control, where the practitioners would like to perform inference with limited amount of data.

## 6  Discussion on failure modes and usage guidelines

From our analysis and experiments, we have identified two major failure modes of SCP: (1) the violation of the causal assumptions, and (2) the failure of the step one CATE estimator. However, the causal assumptions as well as the step one CATE estimation error are *not* empirically verifiable / measurable. Hence, as a practical guideline, we would recommend the analyst to start from the single-cause estimation problem (this is essentially the one-factor-at-a-time principle for scientific discovery [16, 10]).

Suppose the analyst believes that a certain single-cause CATE estimator would perform well in the application. Then, SCP can build upon that estimator to tackle the multi-cause problem (given that the causal assumptions hold). If the analyst believes that no single-cause estimator can perform well in the application, SCP (or other algorithms) are not expected to perform well in the (more challenging) multi-cause setting. If the analyst instantiates SCP with a poorly performing single-cause estimator (Figure 5 B), the procedure may fail due to excessive step one error.

## 7  Conclusion and future works

SCP is a principled way to leverage existing single cause CATE estimation algorithms in the multi-cause setting. It increases sample size and balancing by augmenting the observational dataset with the estimated potential outcomes. In principle, SCP may be used jointly with other data augmentation procedures in the first step to produce an even richer training dataset [67]. Although we make the unconfoundedness assumption in this work, it may also be possible to modify SCP to overcome certain types of hidden confounders [72]. We will leave these extensions to future works.

## Acknowledgments and Disclosure of Funding

We thank anonymous reviewers as well as members of the vanderschaar-lab for many insightful comments and suggestions. This work is supported by the US Office of Naval Research (ONR), the National Science Foundation (NSF Grant number:1722516), and AstraZeneca.

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
