# A  Appendix

**Summary of Appendices.**

Theoretical results and discussions.

- A.1: Proofs and discussions about all propositions.
- A.2: Analysis of CFR in the multi-cause setting.
- A.3: Interpretation of the assumptions.
- A.4: Alternative formalism using do-operation.

Applications.

- A.5: Discussion about SCP's application in medicine

The SCP algorithm.

- A.6: Adapting SCP for different levels of causal structural knowledge.
- A.7: Pseudocode of SCP.
- A.8: Details of the DR-CFR algorithm used by SCP.

Further discussions on related works.

- A.9: Extended related works.

The experiments.

- A.10: Details of the simulation study.
- A.11: Details of the semi-synthetic experiment.
- A.12: Additional simulation results.

### A.1 Theoretical results

**Proposition 1** (Equivalence of the single and multi-cause PO's conditional expectation). *Under the sequential ignorability assumption [53] that $\forall a_k \in \Omega_k$ and $\mathbf{a}_{-k} \in \Omega_{-k}$,*

$$\text{if } \mathbf{A}_{-k}(a_k) = \mathbf{a}_{-k}, \; Y(a_k) = Y(a_k, \mathbf{a}_{-k}). \tag{8}$$

$$Y(a_k, \mathbf{a}_{-k}) \perp\!\!\!\perp \mathbf{A}_{-k}(a_k) | \mathbf{X}, \tag{9}$$

*We have the following equivalence:*

$$\mathbb{P}(Y(a_k, \mathbf{a}_{-k}) | \mathbf{X}) = \mathbb{P}(Y(a_k) | \mathbf{X}, \mathbf{A}_{-k}(a_k) = \mathbf{a}_{-k}) \tag{10}$$

**Proof:** We start by recognizing the right hand side of the Equation 10 follows

$$\begin{aligned}
&\mathbb{P}(Y(a_k) | \mathbf{X}, \mathbf{A}_{-k}(a_k) = \mathbf{a}_{-k}) \\
&= \mathbb{P}(Y(a_k, \mathbf{A}_{-k}(a_k)) | \mathbf{X}, \mathbf{A}_{-k}(a_k) = \mathbf{a}_{-k}) &&\text{by Eq. 8} \\
&= \mathbb{P}(Y(a_k, \mathbf{a}_{-k}) | \mathbf{X}, \mathbf{A}_{-k}(a_k) = \mathbf{a}_{-k}),
\end{aligned}$$

$\forall a_k \in \Omega_k$ and $\mathbf{a}_{-k} \in \Omega_{-k}$, where the second equality holds by Equation 8. Furthermore, by Equation 9, we have for all $\mathbf{a}_{-k} \in \Omega_{-k}$

$$\mathbb{P}(Y(a_k, \mathbf{a}_{-k}) | \mathbf{X}, \mathbf{A}_{-k}(a_k) = \mathbf{a}_{-k}) = \mathbb{P}(Y(a_k, \mathbf{a}_{-k}) | \mathbf{X}).$$

Combining the previous two equations, we immediately see that $\mathbb{P}(Y(a_k, \mathbf{a}_{-k}) | \mathbf{X}) = \mathbb{P}(Y(a_k) | \mathbf{X}, \mathbf{A}_{-k}(a_k) = \mathbf{a}_{-k})$, $\forall a_k \in \Omega_k$ and $\mathbf{a}_{-k} \in \Omega_{-k}$, which concludes the proof.

**Proposition 2** (Identification of the single-cause causal effect). *With the single-cause consistency and unconfoundedness assumptions and the multi-cause overlap assumption (Section 2), i.e.*

$$\forall k \leq K, \; \forall a_k \in \{0, 1\}, \; \text{if } A_k = a_k, \mathbf{Y}'_k(a_k) = \mathbf{Y}'_k \tag{11}$$

$$\mathbf{Y}'_k(a_k) \perp\!\!\!\perp A_k | \mathbf{X}'_k, \; \forall a_k \in \{0, 1\}, \; \forall k \leq K \tag{12}$$

$$\mathbb{P}(\mathbf{A} = \mathbf{a} | \mathbf{X}) > 0, \forall \mathbf{a} \in \Omega, \; \text{if } \mathbb{P}(\mathbf{X}) > 0 \tag{13}$$

*we can estimate the combined single-cause PO as follows:*

$$\mathbb{P}(\mathbf{Y}'_k(a_k) | \mathbf{X}'_k) = \mathbb{P}(\mathbf{A}^{\downarrow}_{-k} | \mathbf{X}'_k, A_k = a_k) \cdot \mathbb{P}(Y | \mathbf{X}'_k, \mathbf{A}^{\downarrow}_{-k}, A_k = a_k). \tag{14}$$

where we denote $\mathbf{X}' := (\mathbf{X}, \mathbf{A}^{\uparrow}_{-k})$ as the combined confounders and $\mathbf{Y}' := (Y, \mathbf{A}^{\downarrow}_{-k})$ as the combined outcomes (Equation 3). The probabilities are evaluated under the conditions $\mathbf{X}' = \mathbf{x}'$, $A_k = a_k$, and $\mathbf{A}^{\downarrow}_{-k}(a_k) = \mathbf{A}^{\downarrow}_{-k} = \mathbf{a}^{\downarrow}_{-k}$.

*Proof.* We start the proof by recognizing that the LHS can be decomposed into two terms by Bayes rule: $\mathbb{P}(\mathbf{Y}'_k(a_k) | \mathbf{X}'_k) = \mathbb{P}(\mathbf{A}^{\downarrow}_{-k}(a_k) | \mathbf{X}') \cdot \mathbb{P}(Y(a_k) | \mathbf{X}', \mathbf{A}^{\downarrow}_{-k}(a_k))$. Hence to prove Proposition 2, we only need to show:

$$\mathbb{P}(\mathbf{A}^{\downarrow}_{-k}(a_k) | \mathbf{X}') = \mathbb{P}(\mathbf{A}^{\downarrow}_{-k} | \mathbf{X}', A_k), \tag{15}$$

$$\mathbb{P}(Y(a_k) | \mathbf{X}', \mathbf{A}^{\downarrow}_{-k}(a_k)) = \mathbb{P}(Y | \mathbf{X}', A_k, \mathbf{A}^{\downarrow}_{-k}), \tag{16}$$

Also note that the single-cause unconfoundedness assumption (Equation 12) implies the following two equations by the properties of conditional independence [12].

$$\mathbf{A}^{\downarrow}_{-k}(a_k) \perp\!\!\!\perp A_k | \mathbf{X}' \tag{17}$$

$$Y(a_k) \perp\!\!\!\perp A_k | \mathbf{X}', \mathbf{A}^{\downarrow}_{-k}(a_k) \tag{18}$$

To prove Equation 15, we treat $\mathbf{A}^{\downarrow}_{-k}$ as the "outcome", $A_k$ as the single cause and $\mathbf{X}'$ as the confounders. We immediately observe that *unconfoundedness* is satisfied by Equation 17. Moreover, *consistency* $\mathbf{A}^{\downarrow}_{-k} = \mathbf{A}^{\downarrow}_{-k}(a_k)$ if $A_k = a_k$ is implied by Equation 11. The multi-cause overlap assumption (Equation 13) implies single-cause *overlap* $\mathbb{P}(A_k | \mathbf{X}') > 0$. Based on these three assumptions, Equation 15 is given by the standard CATE identification theory [56].

Equation 16 can be proved in a similar fashion. Here we treat $Y$ as the outcome, $A_k$ as the single cause and $\left(\mathbf{X}', \mathbf{A}^{\downarrow}_{-k}(a_k)\right)$ as the confounders. From Equation 11 and 18, we observe that the three assumptions are satisfied. Invoking the standard identification theory again, we obtain

$$\mathbb{P}\left(Y(a_k)|\mathbf{X}', \mathbf{A}^{\downarrow}_{-k}(a_k)\right) = \mathbb{P}\left(Y|\mathbf{X}', \mathbf{A}^{\downarrow}_{-k}(a_k), A_k = a_k\right)$$

The consistency assumption 11 states that when $A_k = a_k$, $\mathbf{A}^{\downarrow}_{-k}(a_k) = \mathbf{A}^{\downarrow}_{-k}(A_k) = \mathbf{A}^{\downarrow}_{-k}$. Hence, we can replace $\mathbf{A}^{\downarrow}_{-k}(a_k)$ on the right hand side with $\mathbf{A}^{\downarrow}_{-k}$ to obtain Equation 16, which concludes the proof. $\square$

## A.2 Analysis of CFR in the multi-cause setting

The counterfactual regression method (CFR) was proposed by [57] as a principled way of estimating single-cause CATE. Many methods in the literature extends CFR or uses it as a component. Examples include DR-CFR [22], DragonNet [58], and CEVAE [38]. Given its popularity, we present a theoretical analysis of CFR in the multi-cause setting, generalizing the original analysis presented in [57]. From the analysis, we highlight the challenge of applying CFR in the multi-cause setting. Although the following analysis is for CFR, we believe similar results also hold for the various extensions of CFR mentioned above.

**Review of CFR**. The CFR architecture contains two components: (1) an invertible representation network $\Phi(\cdot) : \mathbb{R}^D \to \mathbb{R}$ that transforms the covariate $\mathbf{X}$ into a real number $r$, and (2) for each treatment $\mathbf{a} \in \Omega$, a prediction head $h_{\mathbf{a}}(\cdot) : \mathbb{R}^D \to \mathbb{R}$ that predicts the PO $Y(\mathbf{a})$ based on representation $r$. Since there are $2^K$ treatments in the multi-cause setting, CFR will include $2^K$ prediction heads. Denote $\Psi$ as the inverse of $\Phi$. Define the factual loss of treatment $\mathbf{a}$ as

$$\epsilon_F(\mathbf{a}) = \int \ell_{\Phi, h_{\mathbf{a}}}(\mathbf{x})\mathbb{P}(\mathbf{X} = \mathbf{x}|\mathbf{A} = \mathbf{a})d\mathbf{x}, \tag{19}$$

where $\ell_{\Phi, h_{\mathbf{a}}}$ is a loss function such as the squared error $(Y(\mathbf{a}) - \hat{Y}(\mathbf{a}))^2$ for treatment $\mathbf{a}$ evaluated at $\mathbf{x}$. Similarly, the counterfactual loss of treatment $\mathbf{a}$ is defined as

$$\epsilon_{CF}(\mathbf{a}) = \int \ell_{\Phi, h_{\mathbf{a}}}(\mathbf{x})\mathbb{P}(\mathbf{X} = \mathbf{x}|\mathbf{A} \neq \mathbf{a})d\mathbf{x}. \tag{20}$$

Note that the counterfactual loss is evaluated using the individuals who did not receive treatment $\mathbf{a}$. The overall counterfactual loss is the average of the treatment-specific counterfactual losses:

$$\epsilon_{CF} = \frac{1}{2^K} \sum_{\mathbf{a} \in \Omega} \epsilon_{CF}(\mathbf{a}). \tag{21}$$

**Proposition 3** (Error bound on the counterfactual loss)**.** *Suppose the function $\ell_{\Phi, h_{\mathbf{a}}}(\Psi(r)) \in G, \forall \mathbf{a} \in \Omega$, where $G$ is a set of functions. The counterfactual loss $\epsilon_{CF}$ is bounded by*

$$\epsilon_{CF} \leq \frac{1}{2^K} \sum_{\mathbf{a} \in \Omega} \epsilon_F(\mathbf{a}) + IPM_G\left(\mathbb{P}(r|\mathbf{A} = \mathbf{a}), \mathbb{P}(r|\mathbf{A} \neq \mathbf{a})\right),$$

*where $IPM_G$ stands for the Integral Probability Metric of function family $G$.*

*Proof.* We start the proof by computing the difference between the treatment-specific factual and counterfactual losses:

$$\epsilon_{CF}(\mathbf{a}) - \epsilon_F(\mathbf{a}) =$$

$$\int \ell_{\Phi, h_{\mathbf{a}}}(\mathbf{x})\left(\mathbb{P}(\mathbf{X} = \mathbf{x}|\mathbf{A} \neq \mathbf{a}) - \mathbb{P}(\mathbf{X} = \mathbf{x}|\mathbf{A} \neq \mathbf{a})\right)d\mathbf{x}$$

$$= \int \ell_{\Phi, h_{\mathbf{a}}}(\Psi(r))\left(\mathbb{P}(r|\mathbf{A} \neq \mathbf{a}) - \mathbb{P}(r|\mathbf{A} \neq \mathbf{a})\right)dr$$

$$\leq \sup_{g \in G} \int g(\Psi(r))|\mathbb{P}(r|\mathbf{A} \neq \mathbf{a}) - \mathbb{P}(r|\mathbf{A} \neq \mathbf{a})|dr$$

$$= IPM_G\left(\mathbb{P}(r|\mathbf{A} = \mathbf{a}), \mathbb{P}(r|\mathbf{A} \neq \mathbf{a})\right),$$

where we (1) apply the definition 19 and 20, (2) use the invertibility of $\Phi$ to change the variable $\mathbf{x}$ into $r$, (3) use the property of the supremum operation and (4) apply the definition of IPM. The proposition can be proved by plugging the above equation into the definition 21 and rearranging the terms. $\qquad\square$

**Comments on Proposition 3**. Proposition 3 demonstrates that balancing distribution via IPM does not scale to the multi-cause setting. In order to minimize the bound on the counterfactual loss $\epsilon_{CF}$, the CFR network needs to find representation $r$'s such that $\sum_{\mathbf{a}\in\Omega} \mathrm{IPM}_G\big(\mathbb{P}(r|\mathbf{A}=\mathbf{a}), \mathbb{P}(r|\mathbf{A}\neq\mathbf{a})\big)$ is small. However, the summation involves $2^K$ terms, each of which involves calculating the IPM between two distributions. Evaluating a single IPM can already be computationally challenging, e.g. it might involve calculating the Wasserstein distance [71], let alone evaluating $2^K$ of them. Moreover, the probabilities $\mathbb{P}(r|\mathbf{A}=\mathbf{a})$ and $\mathbb{P}(r|\mathbf{A}\neq\mathbf{a})$ are not known a priori, but have to be approximated with finite samples. With a large number of treatments, it is questionable whether there will be enough data to approximate these probabilities. Last but not least, even if the IPM terms are well-minimized, it is still challenging to minimize the factual loss $\sum_{\mathbf{a}\in\Omega} \epsilon_F(\mathbf{a})$ because $2^K$ prediction heads $h_{\mathbf{a}}$ need to be trained with finite data. In conclusion, Proposition 3 is a negative finding that discourages us to directly apply distribution balancing via IPM to the multi-cause setting.

### A.3 Interpretation of the assumptions

In this section, we further discuss the various assumptions mentioned in the main text. In particular, we will provide interpretations, examples and additional references.

**The consistency assumption and extensions**. As a reminder, we restate the standard consistency assumption [55] and the extensions we used in the main text.

$$\text{if } \mathbf{A} = \mathbf{a}, \ Y(\mathbf{a}) = Y \qquad\qquad \text{Sec. 2.1}$$
$$\text{if } A_1 = a_1, \ \mathbf{A}_{-1}^{\downarrow} = \mathbf{A}_{-1}^{\downarrow}(a_1) \qquad\qquad \text{Eq. 11}$$
$$\text{if } \mathbf{A}_{-1}(a_1) = \mathbf{a}_{-1}, \ Y(a_1) = Y(a_1, \mathbf{a}_{-1}). \qquad\qquad \text{Eq. 8}$$

In a nutshell, the consistency assumption offers a way to link the observed factual world with the various potential worlds. Under consistency, the factual world is the potential world that corresponds to the observed treatment variable.

While the consistency assumption is widely used in the literature, it may not always hold in practice. A well-known example is the so-called *spill-over effect* [3], where the treatment decision on one patient generates an effect on another patient. In this case, additional adjustments (and assumptions) will be necessary to account for the inconsistency. Our current work focuses on the case where consistency holds and we leave the violation of consistency in the multi-cause setting to future work.

**The unconfoundedness assumption and extensions**. As a reminder, we restate the standard unconfoundedness assumption [24] and the extensions we used in the main text.

$$Y(\mathbf{a}) \perp\!\!\!\perp \mathbf{A} \mid \mathbf{X}, \ \forall \mathbf{a} \in \Omega$$
$$\big(Y(a_1), \mathbf{A}_{-1}^{\downarrow}(a_1)\big) \perp\!\!\!\perp A_1 | \mathbf{X}, \mathbf{A}_{-1}^{\uparrow}, \ \forall a_1 \in \Omega_1. \ \text{Eq. 12}$$
$$Y(a_1, \mathbf{a}_{-1}) \perp\!\!\!\perp \mathbf{A}_{-1}(a_1) | \mathbf{X}, \ \forall a_1 \in \Omega_1. \ \text{Eq. 9}$$

Equation 12 directly mirrors the standard unconfoundedness assumption, where we treat the combined $\big(Y(a_1), \mathbf{A}_{-1}^{\downarrow}(a_1)\big)$ as the "outcome", $A_1$ as the cause and $\big(\mathbf{X}, \mathbf{A}_{-1}^{\uparrow}\big)$ as the confounders.

Equation 9 is a strengthened version of the standard unconfoundedness assumption. Note that this equation concerns with the independence of the potential cause $\mathbf{A}_{-1}(a_1)$ rather than the cause $A_1$. Equation 9 is a standard assumption in estimating the *indirect* causal effect from the mediators [53]. [45] provides ways to verify this assumption when the causal DAG is known.

### A.4 Formalism using do-operation

In the main text, we presented the problem formulation in the potential outcome framework [55]. For completeness, here we present an alternative formalism using do-operation [46]. We highlight that one can derive the same SCP algorithm using *either* formalism.

| Disease | Medications | Reference |
|---|---|---|
| Asthma | Inhaled corticosteroids; Long-acting $\beta_2$-agonists; Leukotriene receptor agonists | [41] |
| Cancer | Dipyrone, Morphine, Fentanyl, Dexamethasone, Metoclopramide | [33] |
| COPD | Inhaled corticosteroids, Inhaled anticholinergics, Diuretics, Proton pump inhibitors | [14] |
| Crohn's disease | IBD drugs, Non-narcotic analgesics, GI symptomatic drugs | [9] |
| Diabetes | Sulfonylureas, Biguanides, $\alpha$-Glucosidase inhibitors, Insulins | [17] |
| Heart failure | ACE inhibitors, ARBs, $\beta$-blockers, Spironolactone | [40] |
| Hypertension | ACE inhibitors, NSAIDs, Acetaminophen, Glucocorticoids, Erythropoietin | [42] |
| Schizophrenia | Antipsychotics, Antidepressants, Benzodiazepines | [66] |

Table 4: Examples of Polypharmacy in complex systemic diseases.

**CATE estimation using do-operation**. Consider the causal graph in Figure 6 (1), where as usual $\mathbf{A}$ represents the causes, $Y$ represents the outcome and $\mathbf{X}$ represents the covariates. The multi-cause CATE can be defined using do-operation [46]:

$$\tau(\mathbf{a}, \mathbf{a}', \mathbf{x}) = \mathbb{E}(Y|\mathbf{X}; do(\mathbf{A} = \mathbf{a})) - \mathbb{E}(Y|\mathbf{X}; do(\mathbf{A} = \mathbf{a}'))$$

where the condition is taken with respect to $\mathbf{X} = \mathbf{x}$. Since the covariates $\mathbf{X}$ satisfies the *backdoor criterion* in the causal graph (Figure 6 (1)), one can express the do-operation using conditional expectation [46]:

$$\mathbb{E}(Y|\mathbf{X}; do(\mathbf{A} = \mathbf{a})) = \mathbb{E}(Y|\mathbf{X}, \mathbf{A} = \mathbf{a}) \tag{22}$$

Note that the conditional expectation in the rhs can be estimated from observed data.

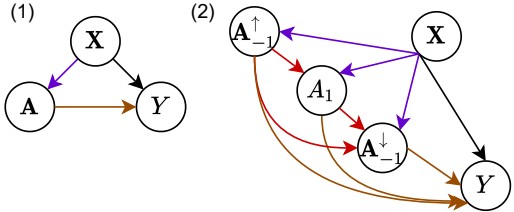

Figure 6: Causal graphs.

**SCP using do-operation**. Now we consider estimating the causal effect of intervention on a single cause $A_1$ (the same analysis holds for any $A_k$ with $k > 1$). We assume that the causal structure between $(A_1, \ldots, A_K)$ is *acyclic*, i.e. the causal graph is a DAG without any loop [46]. Due to the acyclic structure, we can partition the aggregated cause $\mathbf{A}$ into three parts: (1) the single cause $A_1$, (2) the single cause's descendants $\mathbf{A}^{\downarrow}_{-1}$ and (3) its non-descendants $\mathbf{A}^{\uparrow}_{-1}$. The causal graph after partitioning is illustrated in Figure 6 (2). Note that we also allow $\mathbf{A}^{\uparrow}_{-1}$ to directly influence $\mathbf{A}^{\downarrow}_{-1}$ without going through $A_1$. As we can see, an intervention $do(A_1 = a_1)$ will influence both $Y$ and $\mathbf{A}^{\downarrow}_{-1}$:

$$\mathbb{P}(Y, \mathbf{A}^{\downarrow}_{-1}|\mathbf{X}, \mathbf{A}^{\uparrow}_{-1}; do(A_1 = a_1)) = \\ \mathbb{P}(Y, \mathbf{A}^{\downarrow}_{-1}|\mathbf{X}, \mathbf{A}^{\uparrow}_{-1}, A_1 = a_1), \tag{23}$$

where we translate the do-operation (LHS) into a conditional expectation (rhs) because the variables $(\mathbf{X}, \mathbf{A}^{\uparrow}_{-1})$ block the backdoor paths between the "outcome" $(Y, \mathbf{A}^{\downarrow}_{-1})$ and the cause $A_1$. Of course, the conditional probability on the rhs can be estimated in two steps: $\mathbb{P}(\mathbf{A}^{\downarrow}_{-1}|\mathbf{X}, \mathbf{A}^{\uparrow}_{-1}, A_1 = a_1)$ (potential cause) and $\mathbb{P}(Y|\mathbf{X}, \mathbf{A})$ (PO). This mirrors the identification result we derived in Proposition 2 using the PO framework. Moreover, in the second step $\mathbb{P}(Y|\mathbf{X}, \mathbf{A})$ is equal to our ultimate goal $\mathbb{P}(Y|\mathbf{X}, do(\mathbf{A} = \mathbf{a}))$ by Equation 22. Hence, we can see that the augmented dataset generated by SCP is indeed relevant to the goal of multi-cause CATE estimation (similar to Proposition 1).

### A.5 Importance of multi-cause CATE estimation in medicine and beyond

Traditional clinical trials are designed to measure the efficacy and toxicity of a *single* drug. However, in reality a single drug may not be enough to cure the disease. As a result, it is a common clinical

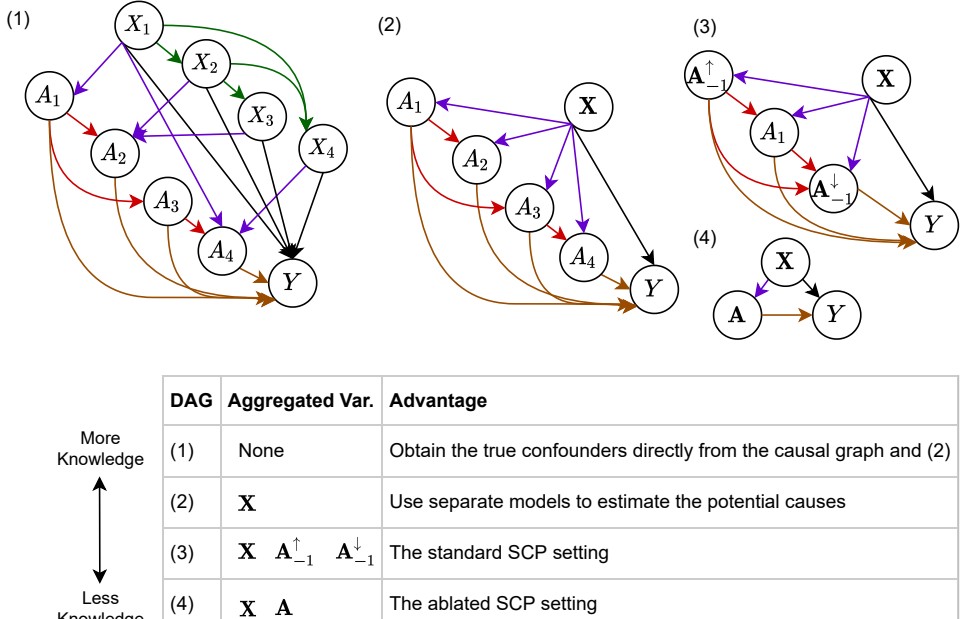

| | DAG | Aggregated Var. | Advantage |
|---|---|---|---|
| More Knowledge | (1) | None | Obtain the true confounders directly from the causal graph and (2) |
| | (2) | $\mathbf{X}$ | Use separate models to estimate the potential causes |
| | (3) | $\mathbf{X}\quad \mathbf{A}^{\uparrow}_{-1}\quad \mathbf{A}^{\downarrow}_{-1}$ | The standard SCP setting |
| Less Knowledge | (4) | $\mathbf{X}\quad \mathbf{A}$ | The ablated SCP setting |

Figure 7: Causal graphs with different levels of structural knowledge. From (1) to (4), the amount of structural knowledge decreases as more variables are aggregated. The standard SCP algorithm operates in setting (3).

practice to prescribe *multiple* drugs together (known as *polypharmacy*, [21]). In order to design the best treatment plan, the clinician needs to know the combined effect of multiple drugs, which is a multi-cause CATE estimation problem.

Polypharmacy is highly prevalent in elderly patients who suffer from multiple chronic diseases. According to several large-scale international studies, around 50% of the elderly population in developed countries take five or more drugs; around 10% take ten or more drugs [18, 6, 39]. Moreover, polypharmacy is also common among patients who suffer from complex systemic diseases such as chronic obstructive pulmonary disease (COPD). These diseases often impact the patient's physiology in complex ways and therefore require multiple medications. In Table 4, we list several examples of such diseases and the medications commonly prescribed together.

Excessive polypharmacy (taking unnecessary medications) increases the financial burden as well as alleviates the risk of adverse effects. According to one highly-cited study [39], nearly 50% of the elderly population take one or more medications that are *not* medically necessary. Excessive polypharmacy is also known to increase the mortality for many common diseases [17, 66, 40]. With a multi-cause CATE estimation algorithm, the medical researchers can better understand the combined treatment effect of multiple drugs — a step towards precision medicine.

Beyond medicine, estimating multi-cause CATE can also be used to support complex decisions in marketing [5, 75], education [19], policy making [61], and more.

**Potential negative impact** Any treatment effect algorithm could be used negatively if the user intentionally chooses to worsen the outcome. This is very unlikely in our case because the intended users of LHM are clinicians.

## A.6   SCP with different levels of causal structural knowledge

The standard SCP algorithm assumes that we can partition the aggregated cause $\mathbf{A}$ into three parts: (1) the single cause $A_1$, (2) the single cause's descendants $\mathbf{A}^{\downarrow}_{-1}$ and (3) its non-descendants $\mathbf{A}^{\uparrow}_{-1}$ (the same analysis holds for any $A_k$ with $k > 1$). This assumption captures the prior knowledge about the causal structure between the variables. In this section, we discuss the scenarios where we have more (or less) knowledge about the causal structure and how we can adapt the standard SCP algorithm to utilize such knowledge.

---
**Algorithm 1** Single cause perturbation.
---
**Input:** Observational dataset $\mathcal{D}_0 = \{\mathbf{x}_i, y_i, \mathbf{a}_i\}_{i\in[N_0]}$, auxiliary regression algorithm $\mathcal{A}$ (here: DR-CFR), final multi-cause PO-estimator $f_\theta$

**for** $k \in [1, K]$ **do**

    Fit $\hat{f}_{\mathbf{A}^\downarrow_{-k}(a_k)}(\cdot)$ with $\mathcal{A}$ to estimate $\mathbf{A}^\downarrow_{-k}(a_k)$ using the observed data $\mathcal{D}_0$ (Eq. 15).

    Fit $\hat{f}_{Y(a_k)}(\cdot)$ with $\mathcal{A}$ to estimate $Y(a_k)$ using the observed data $\mathcal{D}_0$ (Eq. 16)

    Initialize the $k^{\text{th}}$ augmented dataset $\mathcal{D}_k = \varnothing$

    **for** $i \in [1, N_0]$ **do**

        Perturb the $k^{\text{th}}$ cause: $a'_{ik} = 1 - a_{ik}$

        Set $\mathbf{a}^\uparrow_{i,-k}(a'_{ik}) = \mathbf{a}^\uparrow_{i,-k}$ (Eq. 2)

        Set $\mathbf{a}^\downarrow_{i,-k}(a'_{ik}) = \hat{f}_{\mathbf{A}^\downarrow_{-k}(a_k)}(x_i, \mathbf{a}^\uparrow_{i,-k}, a'_{ik})$

        Set $\tilde{y}_i^k := y(a'_{ik}) = \hat{f}_{Y(a_k)}(x_i, \mathbf{a}^\uparrow_{i,-k}, a_{ik}, \mathbf{a}^\downarrow_{i,-k}(a'_{ik}))$

        Combine the causes $\tilde{\mathbf{a}}_i^k := (a'_{ik}, \mathbf{a}_{i,-k}(a'_{ik}))$

        Add new data point $(\mathbf{x}_i, \tilde{y}_i^k, \tilde{\mathbf{a}}_i^k)$ to $\mathcal{D}_k$

    **end for**

**end for**

Obtain the augmented training data $\mathcal{D}^{Tr} = \{\mathcal{D}_k\}_{k\in[0,K]}$

Train $f_\theta$ to estimate $\mathbb{E}[Y|\mathbf{X}, \mathbf{A}]$ using $\mathcal{D}^{Tr}$ (Eq. 1)

**Output:** A trained multi-cause PO predictor $f_\theta$

---

Figure 7 illustrates four settings (1)-(4) with a decreasing amount of structural knowledge. The scenario (3) corresponds to the setting where the standard SCP operates. In comparison, the causal graph in scenario (2) contains the links between all pairs of causes $A_j$, $A_k$. The additional knowledge might enable us to better estimate the *potential cause*. For instance, we can leverage the causal structure to learn separate models for $A_2 = f_2(A_1, \mathbf{X})$, $A_3 = f_3(A_1, \mathbf{X})$ and $A_4 = f_4(A_3, \mathbf{X})$ rather than learning a joint model $(A_2, A_3, A_4) = f(A_1, \mathbf{X})$, which is less parsimonious. However, in practice, our ablation study in Section 5.1 indicates that the performance gain of scenario (2) might be very marginal because the gain is small even when the potential cause is set to be the ground truth.

In scenario (1), we have knowledge about the links between all pairs of variables including the ones between the covariates $\mathbf{X}$. This might enable us to better estimate the outcome $Y$ under SCP. For instance, from the DAG we can see that the covariate $X_1$ alone is enough to block the backdoor path between $Y$ and $A_1$, i.e. it is a true confounder. The remaining covariates $X_2$ to $X_4$ only influence the outcome $Y$ but not the cause $A_1$. Having a smaller set of confounders generally improves the estimation accuracy [13]. In comparison, the standard SCP does not assume such knowledge. As a result, we use DR-CFR to *learn* the true confounders from the data rather than obtaining it directly from the causal graph. Details of DR-CFR is given in Appendix A.8.

Finally, the scenario (4) contains the least structural knowledge. From this causal graph, we cannot identify which variables are causally modulated by a single cause. To apply SCP in this setting, we could assume that *all* other causes are under the influence, i.e. $\mathbf{A}^\downarrow_{-1} = \mathbf{A}_{-1}$. This corresponds to the ablated version we examined in Section 5.1, which performs slightly worse than the standard version.

### A.7  Pseudocode of SCP

The Pseudocode is presented in Algorithm 1.

### A.8  Details of DR-CFR

SCP uses the DR-CFR algorithm to solve the auxiliary tasks [22]. DR-CFR contains the following networks:

- Three representation networks to learn the latent factors $\Gamma(\mathbf{x})$, $\Delta(\mathbf{x})$ and $\Upsilon(\mathbf{x})$.
- Two outcome predictions networks, each for one treatment: $h^0(\Delta(\mathbf{x}), \Upsilon(\mathbf{x}))$, $h^1(\Delta(\mathbf{x}), \Upsilon(\mathbf{x}))$

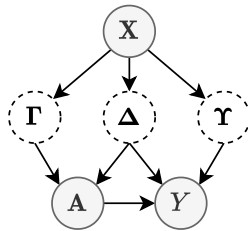

Figure 8: Causal DAG of DR-CFR. Dashed circles denote the latent variables. Shaded circles denote observed variables.

- Two propensity networks to model the probability of treatment assignment: $\pi_0(t|\Gamma(\mathbf{x}), \Delta(\mathbf{x})), \pi(t|\Delta(\mathbf{x}))$

The causal DAG assumed by DR-CFR is presented in Figure 8. The latent factor $\Gamma(\mathbf{x})$ only impacts treatment assignment but not the outcome, whereas the latent factor $\Upsilon(\mathbf{x})$ only impacts the outcomes but not the treatment. The latent factor $\Delta(\mathbf{x})$ is the true confounding factor that both impacts the treatment and the outcomes. To predict $Y(a_i)$, we use the linear layer as the final layer in the outcome prediction network. To predict the binary $\mathbf{A}^{\downarrow}_{-1}(a_1)$ we use the sigmoid function as the final layer. In our implementation, each element in $\mathbf{A}^{\downarrow}_{-1}(a_1)$ corresponds to one output neuron, and all outputs are issued simultaneously. One could alternatively predict the elements in $\mathbf{A}^{\downarrow}_{-1}(a_1)$ one by one to better capture their inter-dependency (e.g. using a recurrent output layer). However, the ablation study in Section 5.1 shows that our current implementation already achieves comparable performance with the Oracle Cause model.

The DR-CFR is trained using the following objective:

$$
\begin{aligned}
\mathcal{J} = & \frac{1}{N} \sum_{i=1}^{N} \omega_i \mathcal{L}\big[y_i, h^{t_i}\big(\Delta(\mathbf{x_i}), \Upsilon(\mathbf{x_i})\big)\big] + \\
& \frac{1}{N} \sum_{i=1}^{N} -\log[\pi_0(t_i|\Gamma(\mathbf{x}_i), \Delta(\mathbf{x}_i))] + \\
& \alpha \mathrm{MMD}\big(\{\Upsilon(\mathbf{x_i})\}_{t_i=0}, \{\Upsilon(\mathbf{x_i})\}_{t_i=1}\big),
\end{aligned}
$$

where $\omega_i$ is the inverse propensity weight, $\mathcal{L}$ is the factual prediction loss (MSE for continuous outcome and cross entropy for binary outcomes), MMD is the maximum mean discrepancy between two samples.

### A.8.1 Why is the standard DR-CFR inadequate for the multi-cause setting?

Remember that the standard DR-CFR attempts to find an outcome specific factor $\Upsilon$ that is unrelated to *any* cause in $\mathbf{A}$. Hence, when there are multiple causes, DR-CFR may struggle to find such factor (it may not even exist). Moreover, the standard DR-CFR uses the multi-head neural network architecture and the IPM regularization that do not scale to large treatment spaces (as we discussed in Appendix A.2)

In contrast, SCP avoids these issues by using DR-CFR in step one for *single-cause* estimation. Here, for each single cause $A_k$ the DR-CFR algorithm may find a different decomposition. In other words, the factors $\Upsilon_k$ $\Gamma_k$ and $\Delta_k$ will be specific to the single cause being considered. This allows the algorithm to exploit the fine-grained causal structure between covariates $\mathbf{X}$ and the individual causes.

Conceptually, while the standard DR-CFR decomposes the *covariates* $\mathbf{X}$, SCP decomposes the *treatment* $\mathbf{A}$ – it exploits the structure between the multiple causes to improve performance.

### A.9 Extended related works.

■ **Mediation analysis (MA).** The traditional goal of MA is to decompose the total effect of a single-cause into various direct and indirect effects [68], and methods generally target population average effects, and not CATE [28]. Conversely, the CATE literature explicitly excludes mediators from

Table 5: Simulation parameters. Bold values are the defaults.

| Parameter | Description | Range |
|---|---|---|
| $K$ | Number of causes | $2, 5, \mathbf{7}, 10$ |
| $D$ | Number of confounders | $10, \mathbf{20}, 30, 40$ |
| $N_0$ | Training sample size | $400, \mathbf{700}, 1400, 2000$ |
| $p_d$ | Sparsity interaction | $\mathbf{0.05}, 0.1, 0.15, 0.2$ |
| $p_s$ | Sparsity single variable | $0.1, \mathbf{0.3}, 0.5, 0.8$ |
| $p_v$ | Sparsity single variable | $0.1, \mathbf{0.3}, 0.5, 0.8$ |
| $\alpha$ | Confounding level | $\mathbf{1}, 3, 5, 7$ |
| sd$(\epsilon)$ | Cause noise | $\mathbf{0.01}, 0.3, 0.5, 0.8$ |
| $\phi(\cdot)$ | Output function | identity, sigmoid |

covariates X; only pre-treatment variables/confounders are considered [69, 23]. Finally, we note that our setting is not the standard setting of mediation analysis; we are not aware of any prior work making a link between multi-cause treatments and mediation analysis.

## A.10  Simulation details.

Table 6: Hyper-parameter values and search ranges.

| Parameter | Value/Range | Algorithm |
|---|---|---|
| Batch size | [50,100,200] | All |
| Hidden dim | Uniform[10,40] | All |
| Confounder rep dim | Uniform[10,40] | DR-CFR, SCP |
| Outcome rep dim | Uniform[10,40] | DR-CFR, SCP |
| IPM weight | [0,1] | DR-CFR, CFR, SCP |
| Learning rate | 0.005 | All |
| Hidden layer | 1 | All |
| Max training epoch | 100 | All |

**Simulation parameter range and the default values.** Table 5 lists the range of simulation parameters with the default values bolded. The confounding level is controlled by scaling the vector $\mathbf{v}$ in Equation 6 by a factor $\alpha$, i.e. $\mathbf{v}' = \alpha \cdot \mathbf{v}$ (bigger $\alpha$, higher confounding).

The results in Figure 5 corresponds to a typical simulation with all settings set to the default. The ones in Figure 4 corresponds to a simulation with $K = 10$ causes and other settings set to the default.

**Hyper-parameter search.** We perform hyper-parameter tuning for all algorithms compared in the experiments. We use randomized search to choose the hyper-parameters that achieve the highest prediction accuracy of the factual outcomes on the validation set. The search range of the hyper-parameters are shown in Table 6.

**Computational resources** The simulations were performed on a server with a Intel(R) Core(TM) i5-8600K CPU @ 3.60GHz and a Nvidia(R) GeForce(TM) RTX 2080 Ti GPU. All individual simulations were finished within 1 hour.

## A.11  Real data experiment details

**Description of the dataset.** We obtained de-identified COVID-19 Hospitalization in England Surveillance System (CHESS) data from Public Health England (PHE) during the first peak of the pandemic, which contains the medical records of 4,714 ICU admissions from 94 NHS trusts across England. The patient covariates $\mathbf{X}$ includes the demographics and the medical conditions prior to the infection. The summary statistics of these covariates are provided in Table 7. The dataset also records the medical treatments administered to the patients. There are five types of medical treatments (causes) in total, and the frequency of each type is summarized in Table 8. The five causes can be combined to give 32 treatment plans in total (e.g. starting with noninvasive ventilation and then switch to invasive ventilation). The frequency of the 32 treatments is plotted in Figure 9. As we can see, many treatments have low frequencies (and hence low propensity scores), which may affect the performance of the propensity weighting methods.

| Covariate | Prevalence | Count |
|---|---|---|
| pneumonia | 25% | 1168 |
| ARDS | 15% | 709 |
| coinfection | 1% | 51 |
| chronic_respiratory | 5% | 248 |
| asthma | 9% | 410 |
| chronic_heart | 5% | 251 |
| chronic_renal | 3% | 132 |
| chronic_liver | 1% | 41 |
| chronic_neurological | 2% | 72 |
| diabetes | 15% | 712 |
| immunosuppression | 3% | 147 |
| obesity | 9% | 432 |
| hypertension | 21% | 969 |
| sex_male | 71% | 3349 |
| age (median) | 58 | - |

Table 7: Summary statistics of the covariates.

| Cause | Percentage | Count |
|---|---|---|
| oxygen via cannulae or mask | 15% | 686 |
| highflow nasal oxygen | 3% | 141 |
| noninvasive ventilation | 16% | 748 |
| invasive mechanical ventilation | 43% | 2026 |
| anticovid19 treatment | 6% | 300 |

Table 8: Summary statistics of the causes.

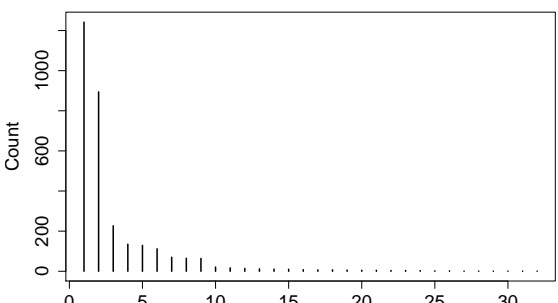

Figure 9: Observed frequency of the 32 treatments.

For SCP, we assume that the five causes do not have direct causal interaction, i.e. $\mathbf{A}^{\downarrow}_{-k} = \varnothing, \forall k \leq K$. In particular, our clinical collaborators belief that the milder form of respiratory support (e.g. oxygen via annulae or mask) does not *causally* modulate the more aggressive form of respiratory support (e.g. invasive mechanical ventilation), and vice versa. The anti-viral treatments are also assumed to be administered independently of the status of respiratory support.

**License and anonymity** Access to the CHESS is regulated. We have signed an end user license before access to the data was granted. All data were pseudonymized in CHESS.

**Hyper-parameter search.** We use the same strategy to tune hyper-parameters as in the simulation study (described in Section A.10).

**Calculation of Spearman's Footrule distance.** The Spearman's Footrule distance measure the disagreement of two rankings [31]. Here the ground truth ranking of the treatments is given by the

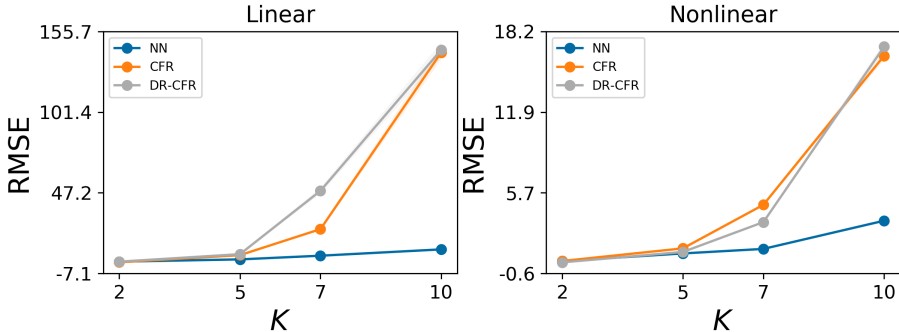

Figure 10: Performance of CFR and DR-CFR significantly degrade as the number of causes $K$ increases. Left panel: the linear setting with identity output function. Right panel: the nonlinear setting with sigmoid output function.

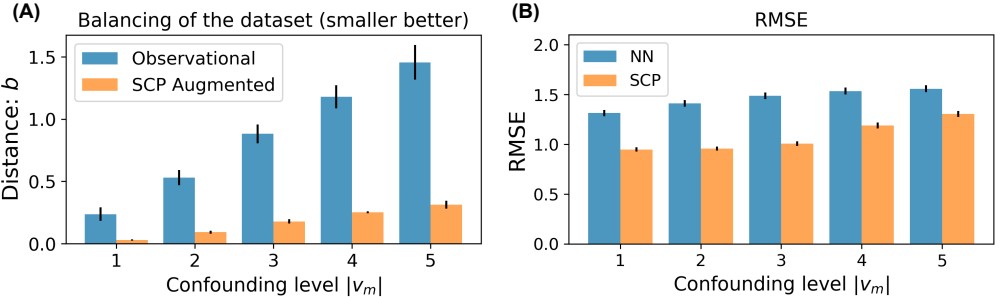

Figure 11: Performance of SCP under different levels of confounding. Left panel: SCP improves the balancing of the observational data. Right panel: the improvements in balancing associates with the reduction in RMSE.

true potential outcomes (treatments with larger POs are ranked higher). Similarly, the predicted ranking is given by the predicted POs.

Given two rankings $\pi$ and $\sigma$, the Spearman's Footrule distance is defined as

$$D(\pi, \sigma) = \sum_{\mathbf{a} \in \Omega} |\pi(\mathbf{a}) - \sigma(\mathbf{a})|$$

In other words, it is the L1-distance between the ranking vectors.

### A.12  Additional simulation results.

#### A.12.1  Standard CFR and DR-CFR under perform in the multi-cause setting

In Figure 10 we present the performance of CFR and DR-CFR under different numbers of causes $K$. These two algorithms are developed for single-cause CATE estimation. As expected, the standard CFR and DR-CFR are not performing well in the multi-cause setting: their performance is even much worse than covariate adjustment (NN) when $K \geq 3$. This is probably because the multi-head architecture and the IPM regularization deteriorate as the number of causes increases (Appendix A.2).

#### A.12.2  Improvements in balancing and reduction in RMSE

We generated a range of observational datasets with varying confounding levels, and use SCP to augment each dataset. Figure 11 (A) shows that SCP's augmented data is consistently more balanced than the observational data (also see Section 5.1). To contextualize the effect of better balancing, we compare the estimation accuracy of the NN trained on the observational data and the one trained on the more balanced augmented data. Figure 11 (B) shows that training on the more balanced augmented data leads to consistent performance improvement across different levels of confounding.

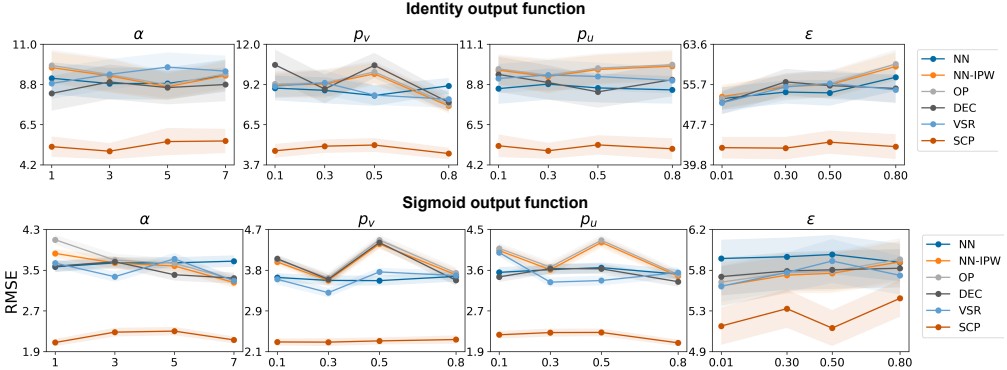

Figure 12: Performance of SCP under different levels of structural sparsity and confounding.

### A.12.3 Simulations with varying structural sparsity and confounding levels

Figure 12 shows the additional simulation results under varying structural sparsity and the confounding levels. Here the structural sparsity is controlled by $p_v$ and $p_u$ (the details were given in Section 5.1). A higher $p_v$ means more covariates will modulate treatment assignment while a higher $p_u$ means the causes will interact with each other more often.

The confounding level is controlled by scaling the vector $\mathbf{v}$ in Equation 6 by a factor $\alpha$, i.e. $\mathbf{v}' = \alpha \cdot \mathbf{v}$ (bigger $\alpha$, higher confounding). The random noise $\epsilon$ involved in the treatment assignment process also modulates the level of confounding (a higher noise level leads to less confounding).

As we can see, SCP outperforms the benchmarks in all cases, often by a significant margin. Its performance is also relatively stable in the various settings.