# OpenReview forum: "Estimating Multi-cause Treatment Effects via Single-cause Perturbation"
_NeurIPS.cc/2021/Conference — NeurIPS 2021 Poster_

### Official Review · Reviewer_YNQn · 2021-06-26

**Rating:** 6
**Confidence:** 3

**Summary:**

In this paper, the authors aim to the estimation of CATE when there are multiple treatment variables and the causal relations between them are known in advance. The main innovation in methodology is an additional step of flipping the value of each variable and updating the corresponding descendants according to the estimated causal mechanisms. This step leads to that more data could be obtained and the data are more balanced, which two steps could benefit the CATE estimation.

**Limitations And Societal Impact:**

Yes.

**Main Review:**

After the rebuttal, the authors have addressed my main questions. I am willing to increase my score.

------
This paper is well written and organized well. The problem the authors tackle is common in reality. And for the notations or concepts that possibly cause ambiguity, the authors provide detailed illustrations. I enjoyed reading this paper.

My main concern is about the novelty. Although I do not see it as a drawback, I correspondingly pay more attention to details. For example, as the discussions on Line 177-181, I am curious about the experimental result if we perturb two variables at once. In that way we could have more data than the current method. I think it will be more persuasive if the authors could provide the support from the theoretical perspective or empirical perspective for the balance between the first step error and the sample size.

In addition, I have one more question. What method did the authors adopt to estimate the first term on the right hand of Eq. 4?

I am willing to increase my score if the authors could provide more details about the balance between the first step error and the sample size, or some further theoretical guarantee for the effectiveness of the proposed method. I also possibly change my score according to other reviewers' comments.

Line 136: It should be ``a, \tilde{a}^1, \ldots, \tilde{a}^K''. If $k$ denotes an index from [K], there should be an ellipsis ahead of \tilde{a}^k.

**Time Spent Reviewing:**

3

---

> ### Author Response · Authors · 2021-08-10
> **Response to Reviewer YNQn**
>
>
> Thank you for your thoughtful comments and suggestions. We’ve provided a point-by-point response below. Please let us know if anything needs further clarification.
>
>
> ---
>
> ### (1) Novelty
>
> The novelty of our work is substantial and multifaceted. To our knowledge, SCP is the first multi-cause CATE estimator that leverages single-cause CATE estimation as an auxiliary task. By bridging the gap between single and multi-cause, SCP can leverage the well-studied single-cause estimators in the more challenging multi-cause setting. Furthermore, the data augmentation approach taken by SCP has not been widely adopted in the CATE estimation literature (despite its success in supervised learning).
>
> ### (2) Analysis of first step error
>
> We have empirically assessed the effect of step one error in Figure 5 (B) and line 266-271. The step one error that is needed to degrade the overall performance turned out to be much bigger than the actual error. In other words, there is a substantial safety margin for the step one estimator.
>
> ### (3) Perturbing two causes at once
>
> We reiterate the two reasons for not perturbing more than one cause in SCP.
>
> 1. *Computational complexity*: perturbing all pairs of causes requires training $K(K-1)/2$ estimators in step one (rather than $K$ estimators). This can significantly increase computational burden (e.g. a five-fold increase when K=10).
>
> 2. *Diminishing returns*: in Figure 4, we presented the results for adding more datasets. We found that the incremental performance gain diminishes as more datasets are added.
>
> We have conducted an additional experiment to further illustrate the diminishing return. We compare the RMSE of the standard SCP and the version that perturbs all pairs of causes (SCP-2). We found the performance gain of SCP-2 to be very marginal for different K’s, which validates our reasoning.
>
> ```
> K=       7             8             9             10
> ===============================================================
> SCP      4.602 (.009)  4.985 (.011)  6.597 (.018)  11.081 (.020)
> SCP-2    4.561 (.008)  4.955 (.011)  6.538 (.017)  11.027 (.020)
> ```
>
> ---
>
> ### Minors
>
> We have used the DR-CFR algorithm as the step one CATE estimator. A self-contained introduction of DR-CFR is provided in Appendix 8. We will correct all the typos in the revision.

---

> > ### Comment · Reviewer_YNQn · 2021-08-18
> > **A clarification for my questions about novelty**
> >
> > Thank you very much for your thorough response! I am sorry that I missed some details in the beginning. Your response addresses some of my concerns. Below, I want to further clarify some of my questions.
> >
> > I agree fully that this paper tackles a new and practical problem. And the idea is interesting and seems not used before in causality literature. They are the advantages. However, considering that perturbation is not a new technique, in addition to the algorithm, I'd like to see the effectiveness of the proposed method (data augmentation by perturbation) from the point of theoretical view. Take a step back, if we ignore this since it is possibly very hard to demonstrate, from the point of algorithm, it is better for the users to know how to select the number of variables we perturb or how to properly achieve the balance between the first step error and overall error \emph{in general}.
> >
> > Above is the main reason that I challenge the novelty. Besides, I agree with the knowledgeable comments by Reviewer sokR. I also possibly adjust my score according to your discussions.
> >
> > Appreciate your response sincerely.
> >
> > Best Regards

---

> > > ### Author Response · Authors · 2021-08-24
> > > **Further response to Reviewer YNQn**
> > >
> > > Thank you for your reply. We fully agree with the reviewer that creating an explicit guideline for when to use the algorithm can be beneficial to the practitioners.
> > >
> > > From our analysis and experiments, we have identified two major failure modes of SCP: (1) the violation of the causal assumptions, and (2) the failure of the step one CATE estimator. However, the causal assumptions as well as the step one CATE estimation error are *not* empirically verifiable / measurable.
> > >
> > > Hence, as a practical guideline, we would recommend the analyst to start from the single-cause estimation problem (this is essentially the one-factor-at-a-time principle for scientific discovery [1-3]).
> > >
> > > 1. Suppose the analyst believes that a certain single-cause CATE estimator would perform well for all causes $A_i$ in the application. Then, SCP can build upon that estimator by perturbing each of the $K$ causes (given that the causal assumptions hold).
> > >
> > > 2. If the analyst is only confident about the estimate for a subset of the causes (e.g. $A_1$ and $A_2$, but not $A_3$), then the data augmentation should perturb these causes alone (e.g. $k=2$).
> > >
> > > 3. If the analyst believes that no single-cause estimator can perform well in the application, SCP (or other algorithms) are not expected to perform well in the (more challenging) multi-cause setting.
> > >
> > > 4. If the analyst does not perform due diligence and instantiates SCP with a poorly performing single-cause estimator, the procedure may fail due to excessive step one error (as we show in Figure 5B).
> > >
> > > ## References
> > >
> > > [1] Fisher, Ronald Aylmer. "Design of experiments." Br Med J 1.3923 (1936): 554-554.
> > >
> > > [2] Hicks, Charles Robert. "Fundamental concepts in the design of experiments." (1964).
> > >
> > > [3] Czitrom, Veronica. "One-factor-at-a-time versus designed experiments." The American Statistician 53.2 (1999): 126-131.

---

> > > > ### Comment · Reviewer_YNQn · 2021-08-26
> > > > **Thank you**
> > > >
> > > > Dear authors:
> > > >
> > > > Thank you for your educational response. I think the current version is sufficient to publish. I am willing to increase the score.
> > > >
> > > > Best Regards

---

> ### Author Response · Authors · 2021-08-16
> **Dear Reviewer YNQn**
>
> Once again, thank you for your invaluable feedback. We were wondering whether our response has addressed your concerns. If you have any additional comments, please let us know, we would be happy to address them.

---

### Official Review · Reviewer_1Xus · 2021-07-16

**Rating:** 8
**Confidence:** 4

**Summary:**

This paper presents a novel method for estimating conditional average treatment effect (CATE) of multiple treatments (or a multi-cause treatment). The research is motivated by data scarcity in multi-cause CATE estimation, and  a single-cause perturbation (SCP) procedure is proposed to augment the observational dataset to improve the estimation.  Detailed theoretical analysis has been provided in the paper (main text and appendices) to support the development of SCP. A comprehensive evaluation of SCP with synthetic data has been conducted, and an application of SCP to a real-world COVID-19 dataset (with simulated outcomes) is presented too. The experiment results have demonstrated the effectiveness and superiority of SCP in comparison with the benchmark methods.

**Limitations And Societal Impact:**

The authors have addressed some specific limitations of their work (e.g. in Section 3.4). It would be ideal if the authors could elaborate a bit more on the restrictions created by the assumptions, e.g. unconfoundedness and sequential ignorability.

**Main Review:**

Please find below the detailed comments on the originality, technical quality, clarity and significance of the paper:
1. Originality:

It is a very interesting and effective idea to make use of single cause CATE estimation for multi-cause CATE estimation. The authors have done a great job in establishing the theoretical link between single cause CATE estimation and multi-cause CATE estimation, and then utilizing this link and the idea of perturbing individual causes to create augmented and more balanced datasets for better multi-cause CATE estimation. The proposed approach has also provided the opportunity to use off-the-shelf single cause CATE estimation algorithms for the multi-cause problem.

From the paper, the originality of the work is clear overall, but to strength the originality further, it would be helpful to clarify the following points:

(a) For single-cause CATE estimation and ATE estimation, is there any existing research which employs data augmentation? If yes, how is the existing research related to SCP proposed in this paper?

(b) Are there any existing research which links single cause CATE or ATE estimation  and multi-cause CATE or ATE estimation? If yes, how is the existing research related to the idea proposed in this paper?

(c) Although in the Appendix, the authors have clarified that the setting of their work is different from mediation analysis, I have a feeling that the analysis in Section 2.2 is related to mediation analysis (or direct vs indirect causal effect), although the overall approach and aim of SCP is different from mediation analysis. It would be good if authors could elaborate on this a bit more.

(d) Related t multi-cause ATE estimation, the joint-IDA method should be mentioned:
Nandy, P., Maathuis, M., & Richardson, T. (2017). ESTIMATING THE EFFECT OF JOINT INTERVENTIONS FROM OBSERVATIONAL DATA IN SPARSE HIGH-DIMENSIONAL SETTINGS. The Annals of Statistics, 45(2), 647-674.

2. (Technical) quality

The proposed method is sound, and detailed discussions (and proofs) have been presented to support the correctness of the SCP algorithm. The experiments are comprehensive and the results are promising. I appreciate the authors' thoroughness in developing, evaluating and presenting the proposed method.

Below are some points that may need clarification to improve the paper:
(a) In terms of partitioning the causes, what about if A_k and some of the other causes are siblings? That is, unlike what's illustrated in Figure 2(B), some other causes are neither A_k's parents/ancestors or children/descendants? I don't think this would change the results the authors have presented, but it would be good to include this case more explicitly.

(b) Lines 120- 121 on page 4: Are the two models trained definitely the predictors for the "potential" treatments or outcome? That is, there is no formal analysis showing that the two factors on the RHS of equation (4) are the potential treatments (A^{\downarrow}_{_k}(a_k)) and the potential outcome (Y(a_k)) respectively. I think you can still get the potential treatments or outcome from data based on (4), but the statement here needs clarification/correction or analysis to support.

(c) Section 3.2: It sounds ok to pool the augmented datasets, but what about if the estimations of the expected outcome (i.e. the RHS of (5) estimated from data) for two different k values are not consistent? Have you tried estimating the RHS for different k values and check their consistency?

(d) Page 7, about the benchmarks: It would be good if more details could be added on the benchmarks, in particular, how those single cause CATE methods are used as benchmarks (i.e. how are they applied for multi-cause CATE estimation)?

3. Clarity/presentation

Although as mentioned above, some technical details might need some clarification/improvement, this paper is well written and very easy to follow. It is enjoyable to read it.

One point though (on Appendix A.1): the sequential ignorability assumption does not seem to appear in reference [50]? Also, it would be good to provide some intuition on this assumption.

4. Significance

Multi-cause CATE estimation is an important and challenging task, and data scarcity is a practical issue for the task. The proposed SCP method could a very useful tool for tackling the challenge. Although strong assumptions are required by SCP, comparing to existing methods (which make similar assumptions), from the experiments, SCP and causal data augmentation look like a promising direction to follow.


**Time Spent Reviewing:**

6

---

> ### Author Response · Authors · 2021-08-10
> **Response to Reviewer 1Xus**
>
> Thank you for your thoughtful comments and suggestions. We’ve provided a point-by-point response below. Please let us know if anything needs further clarification.
>
> ---
>
> ### (1) Data augmentation
>
> Allow us to reiterate, that -- as discussed in Section 4.3 -- to our knowledge, data augmentation has not been used in CATE estimation. We pointed out in Section 4.2 that G-computation for ATE involves sampling multiple datasets for *marginalization* (i.e. to compute an *average* effect). In contrast, SCP’s data augmentation is unrelated to marginalization – its purpose is to increase sample size and balancing for covariate adjustment.
>
> ### (2) Single and multi-cause
>
> As we discussed in the introduction, SCP is the first algorithm that attempts to improve multi-cause CATE estimation by leveraging single-cause estimates. We are unaware of any similar approach for the ATE setting. This is one of the key novelties.
>
> ### (3) Mediation analysis
>
> Indirect causal effects naturally arise in non-trivial causal graphs. This is because the intervention effect on one variable may propagate through the graph via multiple paths. Hence, our analysis in Section 2.2 emcompasses both direct and indirect effects of single causes to ensure the final CATE estimator is valid.
> Unlike Mediation analysis, whose target is to quantify and *decompose* the direct and indirect effects, our goal is to estimate the *total* effect of all causes on the outcome variable (i.e. CATE).
>
> ### (4) Additional citations
>
> We will include the suggested reference [1] in the revision. Note that the key assumption in [1] is that the data is generated by a *linear* structural equation model, which differs from our setting.
>
> ### (5) Siblings
>
> Thank you for raising this point. By definition, the siblings are included in the non-descendants. In fact, one can always partition a DAG based on descendants and non-descendants (which include parents, ancestors and siblings). We will clarify the definition of non-descendants in the revision.
>
> ### (6) Model prediction
>
> On line 120-121: the two single cause models are indeed trained to predict the *potential* outcomes. Equation (4) provides justification for this procedure: the models are trained based on the observable quantities on RHS, and they issue predictions for the potential outcomes on LHS.
>
> ### (7) Pooling datasets
>
> We have conducted an experiment in figure 4, where we start with the original dataset and add the K augmented datasets one-by-one. The performance increases as more datasets are added, which supports the pooling of datasets.
>
> ### (8) Single-cause benchmarks
>
> To apply single cause CATE estimators, we need to aggregate all causes into a single variable -- a one hot vector with 2^K dimensions. We will clarify this in the updated manuscript. As we discussed in the introduction, single-cause CATE estimators are not expected to work well in the multi-cause setting due to combinatorial treatments.
>
> ### (9) Intuition on the assumptions
>
> We will further expand Appendix 3 to provide more intuition about the assumptions. We refer the reviewer to [2] for a thorough discussion about sequential ignorability.
>
> ---
>
> ### References
> [1] Nandy, Preetam, Marloes H. Maathuis, and Thomas S. Richardson. "Estimating the effect of joint interventions from observational data in sparse high-dimensional settings." The Annals of Statistics 45.2 (2017): 647-674.
>
> [2] Forastiere, Laura, Alessandra Mattei, and Peng Ding. "Principal ignorability in mediation analysis: through and beyond sequential ignorability." Biometrika 105.4 (2018): 979-986.

---

> > ### Comment · Reviewer_1Xus · 2021-08-31
> > **Thanks for your response**
> >
> > I'd like to thank the authors for your detailed responses, which have clarified my doubts. I will keep my positive score.

---

### Official Review · Reviewer_yyUp · 2021-07-16

**Rating:** 9
**Confidence:** 4

**Summary:**

The authors propose a data augmentation method for CATE estimation in the setting of multiple causes. In this setting, there is sparsity in the observed treatment assignments, which makes estimation harder. The authors address this sparsity by aggregating multiple datasets together, generated by estimating single cause potential outcomes, which are theoretically shown to be equivalent in expectation to the multi-cause PO's of interest. The method is agnostic to the methods employed for CATE estimation and covariate adjustment on the adjusted data.

**Limitations And Societal Impact:**

Yes

**Main Review:**

**Originality**: Novel method for causal estimation in multi-cause settings. Dispenses with certain assumptions made by past works on multi-cause CATE estimation. In theory, multi-cause methods for ATE estimation could be adapted, but this is discussed. To the best of my knowledge, related work is well cited.

**Quality**: Submission was technically sound and overall of a very high quality. I was particularly impressed by how well the method was explored: discussion of assumptions, possible limitations, avenues for future work outlined, investigation into the inner workings of the method, etc. There is empirical support for the method, as well as theoretical support for why estimating single-cause PO's and using them to augment data is a valid thing to do.

**Clarity**: I thought the paper was very well written and organized.

**Significance**: CATE estimation in multi-cause settings is undoubtedly an important problem. I find this work very compelling because it is a data augmentation scheme that can work in tandem with a wide variety of CATE estimation / covariate adjustment methods, which greatly extends its applicability. The method is scalable but also seems to work well on relatively small datasets. High significance overall.

**Time Spent Reviewing:**

2

---

> ### Author Response · Authors · 2021-08-10
> **Response to Reviewer yyUp**
>
> We are sincerely grateful for your time and energy in the review process.

---

### Official Review · Reviewer_f9co · 2021-07-17

**Rating:** 7
**Confidence:** 4

**Summary:**

This paper proposes a data augmentation approach for conditional average treatment effect (CATE) estimation with multiple treatments. Under the assumption that the multiple treatments follow a DAG, sequential ignorability, and sequential overlap, the conditional average of each (multi-cause) potential outcome can be identified through the conditional average of the corresponding single-cause potential outcome. The proposed method starts by fitting models for (1) how each treatment affects its causal descendants, and (2) how covariates, each treatment and its causal descendants affect the outcome. Then it generates $K$ pseudo-samples, where $K$ is the number of treatments, from the estimated data generating process, each flipping one of the treatment. Finally, it fits a regression model on the augmented data. The proposed method outperforms a handful of competing methods on extensive numerical experiments.

**Limitations And Societal Impact:**

(1) The paper claims that "instead of making additional assumptions on the DGP, we exploit the connection between a single-cause intervention and a multi-cause intervention". This is an overstatement. The proposed method does impose more restrictions on the DGP that is not required in other works (e.g., deconfounder). It requires a DAG structure among the treatment, which is a big assumption. In many applications involving multiple treatments, such as those in two-sided markets, it is unclear whether the acyclic assumption holds among the treatments. Even if it holds, it is not always easy to draw a reliable DAG.

(2) Line 120-123: what does "the model for $A_{-k}^{\downarrow}(a_k)$" and "the model for $Y(a_k)$" mean? In order to generate the pseudo-samples, it seems that they should be the whole distribution instead of just the conditional mean. The algorithm 1 in Appendix does not seem to clarify this.

(3) Line 130-132: similarly, the word "obtain" is confusing. Do you mean sampling from the estimated distribution or plugging in the estimated conditional mean or something else?

(4) Does the proposed method have any connection with the marginal structural models (MSM)?

(5) Line 235: should the metric be $\sqrt{\frac{1}{N_t}\sum_{i=1}^{N_t}\sum_{a\in \Omega}(y_i(a) - \hat{y}_i(a))^2}$? Why do you use $y(a)$ instead of $E[y(a)|x]$?

**Main Review:**

The single-cause perturbation-based data augmentation is novel and interesting for causal inference with complex and high-dimensional treatments. Generally speaking, I feel that data augmentation should be used more in causal inference, because it helps regularizing the estimation under data scarcity. On the other hand, the proposed single-cause perturbation appears to be an effective augmentation scheme because it exploits the DAG structure.

Furthermore, the paper has a very impressive experimental assessment --- it not only shows that the proposed method has a superior performance (i.e., RMSE), but also presents various robustness checks to investigate why it works. For example, Figure 4 and Figure 5 are very informative. They exactly answer some of my questions when I read the previous sections.

**Time Spent Reviewing:**

5

---

> ### Author Response · Authors · 2021-08-10
> **Response to Reviewer f9co**
>
> Thank you for your thoughtful comments and suggestions. We’ve provided a point-by-point response below. Please let us know if anything needs further clarification.
>
> ---
>
>
> ### (1) DAG assumption
>
> Thank you for pointing out that we should revise line 43-44 to avoid an over-statement. Instead of the current formulation, we will highlight that SCP does not assume any parametric outcome model or treatment assignment model, which was the original intention of the statement.
>
> Causal inference with cyclic causal structure is an area of ongoing research [1]. Here we assume the graph is a DAG and will leave the cyclic case for future work.
>
> ### (2) Clarification on pseudo-sample generation
>
> Thank you for pointing out that we should further clarify the pseudo-sample generation procedure, we will discuss the below in the revision.
>
> The pseudo-outcomes are taken as the predicted conditional expectation $\mathbb{E}(Y(a_k) | X_k', A_{-k}^\downarrow)$. This is because the goal is to estimate CATE, which is a conditional expectation. For illustration, consider a linear model $Y = \beta X + \epsilon$, where $\epsilon$ is zero mean noise. Given a dataset with (noiseless) pseudo-outcomes $Y’ = \beta X = E(Y|X)$ and covariates $X$, one can use regression to estimate $E(Y’|X)$, which is in fact equal to $E(Y|X)$. Note that if one needs to quantify the uncertainty around the PO (which is out of the scope of this work), one will need to sample from the full distribution rather than the expectation.
>
> The potential causes $\mathbf{A}^\downarrow_{-k}(a_k)$ need to be sampled from the distribution such that they are still binary variables in the augmented datasets.
>
>  ### (3) Marginal structural models
>
> Marginal structural models (MSM) and SCP are significantly different in the two aspects below. We will include the comparison in the revision.
>
>
> 1.	*Problem setting*. MSM is a class of causal models for identifying treatment effects under *time-varying* exposure [2]. There, the main challenge is that a prior treatment $A_{t-1}$ may causally affect a later covariate $X_{t}$ (Section 7.1 of [1]). In the *static* CATE setting we consider, we only allow the confounder $X$ to affect the treatment $A$ but not vice versa.
> 2.	*Estimation algorithm*.  The parameters of a MSM are usually estimated using the inverse-probability-of-treatment weighted (IPTW) estimators [2], which relates to the various propensity weighting estimators discussed in Section 4.2. In contrast, SCP does not estimate or use propensity score.
>
> ---
>
> ### Minors
>
> We will correct the typo in the equation on line 235.
>
> ### References
> [1] Bongers, S., Peters, J., Schölkopf, B., & Mooij, J. M. (2016). Theoretical aspects of cyclic structural causal models. arXiv preprint arXiv:1611.06221.
>
> [2] Robins, James M., Miguel Angel Hernan, and Babette Brumback. "Marginal structural models and causal inference in epidemiology." (2000): 550-560.

---

> > ### Comment · Reviewer_f9co · 2021-08-23
> > **Reply to the authors**
> >
> > I would like thank the authors for their detailed response. My concerns have been addressed. I will keep my positive score.

---

### Official Review · Reviewer_sokR · 2021-07-21

**Rating:** 7
**Confidence:** 5

**Summary:**

The paper presents a data augmentation method for learning the simultaneous effects of many causes in settings where treatments manipulate many causes at once. Specifically, the authors consider a case where the treatment manipulates a K-dimensional binary vector of variables that cause a single outcome Y. The problem being addressed is sparsity in observed treatments: because there are $2^K$ possible treatments, as $K$ grows, it is less and less realistic that we will observe all levels of the treatment.

The proposed SCP method suggests an approach where the potential outcome distributions for single-cause interventions are learned first, then these are used to generate synthetic data with single-cause perturbations to augment the dataset. The augmented dataset is then used to estimate the potential outcome distributions for all treatments. The authors make several heuristic arguments for why this might work, referencing sample size and covariate balance. They then demonstrate on fully simulated data, and semi-synthetic data using covaraites and treatment indicators for COVID-19 treatment with simulated length-of-stay outcomes.

**Ethical Concerns:**

As noted above, the "real data" example is designed around data from COVID-19 treatment, which demands some care in how those results are presented. As suggested above, this section should be re-titled "semi-synthetic example" to emphasize that the outcomes were simulated.

**Limitations And Societal Impact:**

As stated above, given that this is a method without any theoretical characterization, it is important to highlight cases where the method would fail. As of now, there are only two simulated settings where the method works well; I would encourage the authors to do more stress testing. The ablation study is a useful first step.

**Main Review:**

**Per discussion below, authors' addition of failure cases and general guidance, as well as agreement to switch some framing has moved my recommendation to "Accept"**

# General Feedback

The paper is written clearly, and the problem is well-introduced. However, I remain skeptical of the approach because it is not clear where the information about sparsely-observed treatment combinations is coming from. Because there is no theoretical treatment in the paper or even demonstrations on simple, transparent examples, it is difficult for me to endorse the approach as presented. However, I do think there are some interesting possibilities here that the authors should do more to explore. In my opinion, the current "reasons" for SCP working that are considered are not sufficient to justify the approach, and there are more interesting potential justifications that would be more compelling.

The problem of combinatorially exploding cause combinations has a long history in the statistical experimental design literature, going back to Fisher (see, e.g., https://digital.library.adelaide.edu.au/dspace/bitstream/2440/15191/1/48.pdf). The main issue with so-called factorial experiments is figuring out how to estimate the effects of interactions between causes. For this reason, analysts will often make assumptions about certain orders of interaction being zero (e.g., assuming that there are no 3-way or higher interactions between causes). There are then experimental designs (so-called fractional factorial designs) that can identify the remaining low-order interaction effects (see, e.g., Box, Hunter, and Hunter Chapter 6, hosted here: http://pages.stat.wisc.edu/~yxu/Teaching/16%20spring%20Stat602/%5BGeorge_E._P._Box,_J._Stuart_Hunter,_William_G._Hu(BookZZ.org).pdf).
An extreme case of this is assuming that on a certain scale there are no interactions between causes (e.g., the expectation of the treatment is just a linear combination of the causes). In this case, one could estimate the potential outcome distribution for all 2^K treatments by simply observing K treatment levels, so long as those treatment levels are linearly independent of each other.

This paper seems to aim at addressing a similar issue (at least per the introduction), but now in a more challenging observational context. However, it is not clear how this method addresses the fundamental question of having an exploding number of interaction effects to estimate. The empirical exploration is no help here. Note, for example, that the synthetic experiment only includes first-order effects of causes (there are no interactions between causes, only between causes and covariates), and the simulated outcomes in the "real data" experiment only include, at maximum, second-order interactions between causes. If we knew at the outset that there were only second-order interactions between causes, then we could estimate the effects of all treatment levels with many fewer observed treatment levels (see https://link.springer.com/chapter/10.1007%2Fb105081_8). Thus, SCP could succeed on these experiments by simply implicitly regularizing the estimated treatment effects toward low-order interactions between causes. This is not necessarily a bad thing, but is very different from the current framing of the paper that repeatedly emphasizes that SCP makes no assumptions about the DGP.

There are some ways that I could imagine SCP could help. For example, in cases where there is non-trivial causal structure between the causes, SCP might enforce certain marginalization constraints between causal effects, i.e., that the causal effects of downstream variables should average out in a particular way to match the causal effect of intervening on an upstream variable. These kind of moment constraints could be helpful regularizers for CATE estimators.

There is also a possibility that SCP does nothing that is actually causally interesting, but applies useful regularization to the neural networks used to estimate P(Y | X, A) by self-distillation (see, e.g., https://arxiv.org/abs/2002.05715).

To make their analysis of SCP clearer, I would suggest that the authors examine the method in the following contexts:
* Consider an ideal experimental setting where there is no confounding, and we are just interested in the average treatment effect at each level $\bm a$. This separates out the treatment sparsity problem (the main motivation of this work) from the confounder adjustment problem.
* In the experimental context, show how SCP would work with a simple model that is not a neural network (e.g., a linear model with higher-order interactions with K relatively small). A transparent toy example worked out analytically would be extremely useful, even if it does not prove general properties of the approach. If SCP does very little here, it would suggest that there is something special about the self-distillation.
 * Consider a range of settings with different orders of important interactions. I suspect that SCP will perform worse when higher-order interactions play a major role.
 * Consider a range of cases where the causal structure between the causes is changed systematically. It may be the case that SCP performs better when there is detailed causal structure, as this could correspond to a system of moment constraints that regularize the CATE estimator.

Importantly, when suggesting a causal procedure that relies on unverifiable assumptions, or for which there is no general theoretical justification, it is important to characterize cases where the method breaks, as well as those cases where it succeeds. I suspect that the above analyses would reveal such cases, and this is fine. There is no free lunch when it comes to sparsely observed data, so the method must have limitations. The authors should articulate them clearly.

I consider these to be more essential explantations for SCP's performance than the "balance" and "sample size" considerations listed in the paper, because these would also be reasons that SCP would help with a single binary treatment; they are not unique to the multi-cause context. (As an aside, it would be interesting to see whether SCP also seems to help with estimation in the standard setting. If it does help, then that would again suggest an alternative explanation for why SCP might help with many treatments as well.)

# Detailed Comments

## Notes on Related Work

In Table 2, G-computation and covariate adjustment should have their intermediate estimands listed as P(Y | A, X) since they both both estimate outcome models as nuisance functions. G-computation is also usually done by simply taking the empirical measure of X to be P(X) and P(Y | A, X) is just averaged over the observed values of X. Deconfounder methods generally do not rely on weighting, but instead involve an outcome model P(Y | A, Z).

## Careful with the "Real Data" Example
Section 5.2 uses a dataset with simulated outcomes, so should not be called a real data experiment, especially for such a sensitive topic (COVID-19 Treatment). This should be called a "semi-synthetic" experiment.

**Time Spent Reviewing:**

3 hours

---

> ### Author Response · Authors · 2021-08-10
> **Response to Reviewer sokR**
>
> We would like to thank the reviewer sokR for the extensive review, the suggestions and proposed avenues for future investigations! We greatly appreciate the feedback, and we will include the findings as well as clarifications outlined below in the updated manuscript.
>
> ---
>
> ### General response
>
> Allow us to emphasize that the objective of this work is not to deal with combinatorially exploding treatments in *randomized controlled trials* (RCT). As pointed out in the review, a separate body of work already exists that contains solutions for the RCT setting.
>
> Instead, we focus on the unique challenge arising due to the *interplay* between the observational setting (confounding) and the combinatorial treatments.
> Confounding leads to *distribution shifts* between the treatment groups, which challenges CATE estimation ($P(\mathbf{X}|\mathbf{A}=\mathbf{a}_i) \ne P(\mathbf{X}|\mathbf{A}=\mathbf{a}_j)$, $\forall i\ne j$). Many methods exist to correct the distribution shift between *two* treatment groups, but they often fail to scale with the *combinatorial* treatments (see the references on line 192).
>
> SCP mitigates distribution shift via explicit data augmentation. It creates additional data points by perturbing every single cause *uniformly* and estimating the potential outcomes. The reduction in distribution shift has been validated experimentally (Figure 5).
>
> Consequently, the main failure mode of SCP would be that the imputed potential outcomes differ too much from the true ones (high step one error, Section 3.4). This would be the case whenever (i) the first stage estimator cannot model the expected single-cause outcome (given other fixed causes) and (ii) the single-cause (marginal) distribution shift is not adjusted for.
>
> Because SCP is compatible with *any* off-the-shelf single-cause CATE estimators, it can adopt methods that are state-of-the-art for (i) and (ii). This allows us to use recently proposed algorithms that employ balancing representation (e.g. DR-CFR) to achieve better estimation accuracy (lower step one error).
> If domain knowledge exists, SCP also allows us to use an application-appropriate first stage model which encodes the known structure (e.g. a linear model with pairwise interaction).
> In general, as long as the first stage CATE estimator achieves reasonable error, we expect SCP to perform well due to the reduction in distribution shift.
>
> To contextualize the discussion above, we now show a list of situations where SCP is expected to work (or not to work).
>
> ###  (1) Will SCP achieve performance gain under *no confounding* (e.g. randomized controlled trial)?
>
> **No.** SCP is not expected (or designed) to achieve performance gain in the RCT setting.
> This is because the augmented dataset does not reduce the distribution shift in the original data (there is no distribution shift to start with).
>
> We performed a simulation with no confounder using the same data generation method and parameters in Section 5.1 ($\mathbf{X}$ is removed from both the treatment assignment and the outcome model).
> The results below verify our expectation (K is the number of causes, NN denotes regression with neural networks, the metric is RMSE).
>
> ```
> K=  2             5             8             10
> ==========================================================
> NN  0.027 (.017)  0.085 (.018)  3.461 (.285)  5.327 (.190)
> SCP 0.024 (.013)  0.082 (.017)  2.914 (.248)  5.534 (.208)
> ```
>
> ###  (2) Will SCP achieve performance gain when there are *high-order interactions*?
>
> **Yes.** We reiterate that SCP does *not* assume any parametric outcome model, e.g. a linear model with low-order interaction. SCP is not restricted to low-order interactions.
>
> To further this point, we would like to emphasize that the causes will naturally interact when the outcome model is *nonlinear*. For example, suppose $Y=\sigma(X_1 + X_2) + \epsilon$ where $\sigma$ is the logistic function. The effect of $X_1$ on $Y$ will be smaller when $X_2$ is big (due to the saturation of the logistic function), which indicates an interaction between $X_1$ and $X_2$.
>
> In both synthetic and semi-synthetic experiments (Equation 6 and 7), we have studied the performance of SCP under nonlinear outcome models (with implicit high-order interactions). We have considered the logistic and the exponential nonlinearity. SCP consistently outperforms the benchmarks in these settings (Figure 3 second row, Table 3).
>
> ###  (3) Will SCP achieve performance gain when *linear estimators* are used (instead of DR-CFR and NN)?
>
> **Yes.** Our original reasoning on improving balancing and adjusting distribution shifts still applies.
>
> To show this, we instantiate a linear SCP (SCP-LM): we use a linear regression model with propensity adjustment as the step one CATE estimator, and a standard linear regression model in step two.
>
> The results below show that SCP-LM out-performs a standard linear regression model (LM). The data are generated as described in Section 5.1 with parameters listed in Table 5. (Note the standard deviations are low because LM is a high-bias low-variance estimator compared with NN).
>
> ```
> K=       2             5             8             10
> ===============================================================
> LM       0.696 (.002)  0.766 (.005)  5.168 (.012)  11.250 (.022)
> SCP-LM   0.516 (.002)  0.720 (.006)  4.985 (.011)  11.081 (.020)
> ```
>
> ###  (4) What if *neural network regression* is used in step one instead of a CATE estimator (DR-CFR)?
>
> We expect performance loss compared with standard SCP. This is because the step one error would increase as the algorithm no longer tackles distribution shifts by learning balancing representations.
>
> The simulation result below illustrates this point (SCP-NN denotes SCP with neural network regression in step one, NN denotes neural network regression with observational data).
>
> ```
> K=       2             5             8             10
> ===============================================================
> NN       0.687 (.012)  1.622 (.056)  3.028 (.069)  5.317 (.147)
> SCP-NN   0.678 (.012)  1.598 (.044)  3.023 (.119)  5.461 (.203)
> SCP      0.620 (.013)  1.551 (.072)  2.510 (.082)  4.441 (.133)
> ```
>
> ###  (5) Is *self-distillation* a key reason for improvements?
>
> Thank you for pointing us to the self-distillation literature. The answer to this question is **‘no’**. As self-distillation is only known to work with deep neural networks, it cannot explain the performance gain of SCP with linear estimators (see the section above on linear estimators).
> In addition, we do not see a performance gain when neural network regression is used in step one instead of a CATE estimator, which indicates no significant self-distilling effect (compare NN to SCP-NN in the section above).
>
> ---
>
> ###  Minors
>
> We will incorporate the suggested minor changes in the revision. We will rename Section 5.2 to highlight the dataset is semi-synthetic.

---

> > ### Comment · Reviewer_sokR · 2021-08-17
> > **Remaining Issues**
> >
> > I'd like to thank the authors for their thorough response. The point about managing covariate balance across many different treatment conditions is a useful framing. However, there are still a few issues that remain for me here:
> >
> > ## Framing in terms of sparse observations
> >
> > The paper itself motivates the method as solving the problem of sparsely observations of different treatment combinations, which is a very different problem from covariate imbalance. Sparse observations can also occur in randomized experiments, which is why I focused on it in my review. However, if the main issue being addressed in the paper is actually the issue of covariates being imbalanced across many different treatment combinations, this should be presented as the problem being solved. As of now this feels like a bait-and-switch; if balance is indeed the main problem the method is designed to address, then the intro should probably be rewritten to focus on that.
> >
> > ## Regularization, Interactions, and Explicit Guidelines
> >
> > I will try to be more explicit about why I am so concerned with higher-order interactions. The goal here was not to poke holes in the method, but to try to get the authors to provide more explicit guidelines for usage of SCP. Unlike other ML methods, where practitioners can validate a method on an application-by-application basis, methods for causal effect estimation can't be explicitly validated. Thus, it is essential to provide guidance for cases where the method is likely to work better or worse, based on assumptions about the true data generating process and the properties of the base model being used. I think this is largely missing from the paper.
> >
> > My hunch is that the implicit regularization imposed by SCP would improve estimation when most effects are from main or low-order interaction effects, but would be worse if there are strong higher-order interactions. This may or may not be correct, but I provided it as a potentially salient axis along which the authors might show the effect of SCP and build some useful guidelines for usage.
> >
> > As presented, the authors seem to imply that SCP will work better than using a base model alone in all observational cases using any single-cause CATE method. This seems implausible to me, because the specific form of data augmentation done by SCP seems to operate more as a regularizer rather than adding new information like data augmentation in other ML contexts. To see this, note that when augmenting images, we are adding the additional information that the transformation of an image does not affect its label, which may contradict what a base model would learn without the augmentation. Here, the augmented data simply reflects back what a base model would learn from the same data (including the bias induced by regularization in the base models); thus, the method seems to operates as a form of regularization rather than a genuine injection of new information. Every regularization induces its own bias-variance tradeoff, so there must also be a tradeoff here. Perhaps the authors have a counterargument re: tradeoffs, but without theoretical support, it seems difficult to argue that SCP will always work. If it doesn't always work, that is fine, but practitioners need to know when it will fail. Again, this sort of detail is, in my opinion, necessary for causal inference because practitioners can't validate approaches on a case-by-case basis.
> >
> > Here, it seems that an argument about how the regularization bias interacts with SCP is important. For treatment combinations that are sparsely observed, the vast majority of datapoints in the augmented dataset will come from the augmentation, not from the original data. The predictions used to construct the augmented dataset will be extrapolations, and thus be very sensitive to the bias-variance tradeoff made in the step-one model. In particular, the bias imposed on the step-one model will show up as **systematic** error for the predicted outcomes in the augmented dataset (unlike the centered, random error used in the step one error analysis in line 266). Unfortunately, it would not be possible to assess the bias in these step-one extrapolations in practice.
> >
> > As an example of why this would likely affect higher-order interaction effects, consider that if a CATE estimation technique tends to regularize away differences between the treated and control group covariates (as DR-CFR seems to do), and non-descendent causes are treated as covariates by the single-cause CATE estimator, then the effects of interactions between non-descendent causes and the focal single-cause will likely be understated, so this regularizes these interaction effects toward zero. In a data-generating process that only has low-order interactions (as is the case in all experiments in the paper), this regularization works well. If there are important higher-order interactions, then the bias induced by this regularization would lead to worse performance. This is why I asked for experiments that consider such interactions.
> >
> > The interactions induced by non-linearities are not really an adequate probe; such interactions are very weak and smooth, and when causes are independently distributed, will tend to average out.
> >
> > Notably, it seems that amplifying stage-one regularization seems like a far more compelling explanation of improvement from SCP than improving balance. Consider the results from (4). NN is a valid (if inefficient) CATE estimator, since it is a regression that includes all confounders, and SCP-NN improves balance, but SCP-NN does not improve performance compared to NN. However, the standard SCP method seems to amplify the regularization imposed by DR-CFR when that method is used for stage one.
> >
> > I might raise my rating if the authors were able to show how SCP with DR-CFR behaves in cases where there are non-trivial higher-order interactions. It wouldn't necessarily need to perform better. It's just very important to be able to discuss when the implicit regularization induced by SCP would be expected to work, and perhaps more importantly, fail.

---

> > > ### Author Response · Authors · 2021-08-24
> > > **Further response to Reviewer sokR**
> > >
> > > Thank you for your reply. We have provided a further response below.
> > >
> > > ## Introduction
> > >
> > > In the introduction, we pointed out that many existing balancing and regularization techniques developed for the single-cause observational setting are inadequate for the multi-cause setting because of the enlarged treatment space (line 38-42) -- this is the motivation for developing SCP. Based on your comments, we will make this motivation more explicit and contrast with the works in the randomized controlled trial setting, which focus on designing or selecting a parametric outcome model.
> > > ## Additional information and data augmentation
> > >
> > > SCP is actually injecting additional information in the augmented data. Here, the additional information are the causal assumptions. For example, the no hidden confounder assumption and the causal DAG (Section 2.2).
> > > These assumptions encode the independence structure in the data generating process.
> > > Importantly, they allow the augmented data to reflect the causal effect of an intervention rather than some spurious correlations.
> > > We have shown experimentally that violation of these assumptions (e.g. using an incorrect causal DAG) leads to performance loss (Figure 4).
> > >
> > > From this perspective, the data augmentation used in computer vision is similar to SCP. Let $X$ be the image, $Y$ be the label and $\theta$ be the amount of rotation applied to the image. The image augmentation encodes an *independence* property: $Y\perp \theta | X$, i.e. the rotation does not change the label given the image. Furthermore, the augmented image is in fact a *counterfactual* image under an intervention on the rotation variable $\theta$ (Here $\theta$ is a causal parent of $X$ [1]).
> > >
> > > A key difference between SCP and image augmentation is whether the counterfactuals can be generated *without error*. In computer vision, the rotated images can be accurately obtained. However, in SCP, the effect of a single-cause intervention has to be *estimated* by the step one estimator. Hence, the SCP augmentation suffers from an extra source of error -- the step one estimation error, which we have highlighted in Section 3.4. We have shown that the Oracle SCP can achieve much better performance than SCP due to the elimination of step one error (Figure 4).
> > >
> > > ## Additional experiments
> > >
> > > Below we present the results of two additional experiments, both of which are designed to further probe the failure mode. These findings will be included in the revision.
> > >
> > > **Performance under explicit high-order interaction**
> > >
> > > As suggested by the reviewer, we specify the true outcome model to have three-order interaction (i.e. Equation 6 with additional interactions terms).  The coefficients are drawn i.i.d. from the standard Normal distribution.
> > >
> > > The result below shows that SCP with DR-CFR is not performing worse than neural network regression (NN) in the presence of three-order interaction.
> > >
> > > ```
> > > K=         2             5             8             10
> > > =================================================================
> > > NN         17.99 (.456)  57.56 (1.18)  218.8 (5.10)  457.4 (8.04)
> > > SCP-DRCFR  17.69 (.446)  51.95 (1.20)  183.7 (3.47)  402.8 (8.47)
> > >
> > > ```
> > > We have also studied a four-order interaction model. The results below also indicate no performance loss.
> > >
> > > ```
> > > K=         4             6             8
> > > ===================================================
> > > NN         109.3 (3.45)  245.9 (2.69)  604.1 (12.2)
> > > SCP-DRCFR  103.6 (2.88)  226.9 (5.56)  595.7 (16.7)
> > >
> > > ```
> > >
> > >
> > >
> > >
> > > **Performance when the single-cause estimator’s assumption is violated**
> > >
> > > Here we study a setting where the assumption of DR-CFR is violated. We let every single covariate $X_i$ be a true confounder (they are connected to all causes and the outcome) and the causes are also densely connected ($A_i \rightarrow A_j, \forall i<j$).
> > > In this setting, the outcome factor $\Upsilon$ and the treatment factor $\Gamma$ do not exist (appendix 8).
> > > Hence, DR-CFR’s three-way decomposition will fail (Figure 8), which leads to high step one error. The results below show that SCP with DR-CFR indeed fails in this setting.
> > >
> > > ```
> > > K=       2             5             8             10
> > > ===============================================================
> > > NN       3.239 (.073)  7.418 (.130)  13.26 (.273)  35.44 (.406)
> > > SCP      4.109 (.112)  7.296 (.224)  13.32 (.268)  39.12 (.508)
> > > ```
> > >
> > > ## Guidelines
> > >
> > > We fully agree with the reviewer that understanding the failure mode of a causal inference algorithm is crucial. In the revision, we will expand the discussion and further highlight that SCP is *not* expected to work under all circumstances.
> > >
> > > From our analysis and experiments, we have identified two major failure modes of SCP: (1) the violation of the causal assumptions, and (2) the failure of the step one CATE estimator. However, the causal assumptions as well as the step one CATE estimation error are *not* empirically verifiable / measurable.
> > >
> > > Hence, as a practical guideline, we would recommend the analyst to start from the *single-cause* estimation problem (this is essentially the one-factor-at-a-time principle for scientific discovery [2-4]).
> > >
> > > Suppose the analyst believes that a certain single-cause CATE estimator would perform well in the application. Then, SCP can build upon that estimator to tackle the multi-cause problem (given that the causal assumptions hold).
> > >
> > > If the analyst believes that no single-cause estimator can perform well in the application, SCP (or other algorithms) are not expected to perform well in the (more challenging) multi-cause setting.
> > >
> > > If the analyst does not perform due diligence and instantiates SCP with a poorly performing single-cause estimator (as in the last experiment), the procedure may fail due to excessive step one error.
> > >
> > >
> > > ## References
> > >
> > > [1] Locatello, Francesco, et al. "Weakly-supervised disentanglement without compromises." International Conference on Machine Learning. PMLR, 2020.
> > >
> > > [2] Fisher, Ronald Aylmer. "Design of experiments." Br Med J 1.3923 (1936): 554-554.
> > >
> > > [3] Hicks, Charles Robert. "Fundamental concepts in the design of experiments." (1964).
> > >
> > > [4] Czitrom, Veronica. "One-factor-at-a-time versus designed experiments." The American Statistician 53.2 (1999): 126-131.

---

> > > > ### Comment · Reviewer_sokR · 2021-08-24
> > > > **Thanks for the additional discussion; raising score**
> > > >
> > > > I'd like to thank the authors for their thorough engagement with my comments. In particular, I'm pleased that even though my hypothesis about higher order interactions did not pan out, the authors found other cases where the boundary between success and failure could be reasoned about.
> > > >
> > > > With the demonstration of how violations of the step one model assumptions can cascade into worse performance under SCP, in addition to the fleshed-out guidelines, I feel the paper is much stronger. I am raising my rating to accept.

---

> > > > > ### Author Response · Authors · 2021-08-24
> > > > > **Thank you**
> > > > >
> > > > > We would like to thank the reviewer again for the thoughtful reviews and suggestions. We feel privileged to be a part of such an insightful intellectual discussion.

---

> ### Author Response · Authors · 2021-08-16
> **Dear Reviwer sokR**
>
> Once again, thank you for your invaluable feedback. We were wondering whether our response has addressed your concerns. If you have any additional comments, please let us know, we would be happy to address them.

---

### Decision · Program_Chairs · 2021-09-27

**Decision:**

Accept (Poster)

**Comment:**

The authors propose a data augmentation method for learning conditional average treatment effects with multiple treatments. Assuming sequential ignorability and sequential overlap, the authors propose to first fit models that describe how changing a single cause affects causal descendants; then to augment the data set with new observations; and finally to fit a regression model on the augmented data. This is a strong paper. All reviewers agree that it should be accepted. Reviewer yyUp praises how well the method was explored, including discussion of limitations. Reviewer f9co states that the paper has "a very impressive experimental assessment" that not only shows superiority to competing methods but also investigates why the method works. Reviewer 1Xus states that the paper is “enjoyable to read”. Reviewers 1Xus, sokR and YNQn had some reservations about the method, in particular how well it performs under higher-order interactions. The authors have provided thorough responses, with additional simulations and high-level guidelines to understand failure modes. The authors state that they will expand the discussion with the additional insights.